# ONLINE BUDGET-AWARE GUIDANCE FOR BLOCKWISE DISCRETE TEXT DIFFUSION

## ABSTRACT

Diffusion language models (DLMs) are emerging as strong alternatives to autoregressive decoders, combining iterative refinement, global context access, and natural controllability. Yet controllable text diffusion remains underexplored: unlike vision, where plug-and-play guidance is central, existing methods for text rely on fixed or decaying step-size schedules that are brittle across prompts, constraints, and remasking strategies. The problem is amplified in modern *blockwise* DLMs, where gradient volatility and coverage fluctuate across spans, making schedule-based heuristics unstable. We present **BALE** (Budget-Aware Logit Editing), a plug-and-play controller that replaces heuristic schedules with an online, budget-aware rule. At each step, BALE fuses multi-constraint gradients and selects guidance strength by balancing a residual-targeted movement rule, stability caps based on an EMA Lipschitz proxy and a KL surrogate. Beyond per-step control, BALE allocates a fixed sequence-level budget across blocks and reallocates dynamically at boundaries. A discrete-time KL bound and a continuous-time SDE view explain why BALE stabilizes blockwise decoding. On LLaDa-8B, BALE improves lexical (must-have, forbidden), semantic (sentiment, formality), and combined controls over prompt tuning and schedule-based baselines, yielding higher satisfaction rates at matched or lower perplexity and stronger human/LLM preference. BALE is lightweight, scheduler-free, and advances controllable text diffusion toward practical deployment.

## 1 INTRODUCTION

Diffusion language models (DLMs) (He et al., 2022; Ye et al., 2024b) have recently emerged as a competitive alternative to autoregressive (AR) decoders for long-form text generation. By iteratively denoising masked spans, they enable error correction across steps, offer better exposure to global context, and naturally expose interfaces for controllable sampling (Ye et al., 2025; Nie et al., 2025). In vision, controllability has become one of the defining advantages of diffusion models, with extensive work on classifier guidance, classifier-free guidance, and training-free guidance mechanisms (Dhariwal & Nichol, 2021; Ho & Salimans, 2022; Ye et al., 2024a). These techniques allow plug-and-play steering of attributes without retraining and have been central to the rapid adoption of diffusion in generative imaging.

In contrast, controllable text diffusion remains relatively underexplored. Early work such as Diffusion-LM (Li et al., 2022) demonstrated that discrete diffusion can improve controllable generation, but relied on fixed or linearly decaying schedules for the guidance scale, which are brittle across prompts and tasks. DiffuSeq (Gong et al., 2022) extended diffusion to conditional sequence-to-sequence tasks, but did not provide general-purpose controllability at decoding time. More recent efforts like SSD-LM (Han et al., 2022) introduced modular control heads, yet these still depend on retraining or task-specific settings rather than a flexible plug-and-play interface. Training-free guidance for discrete diffusion has also been proposed (Kerby & Moon, 2024), but evaluations have focused on small-scale models and toy settings, leaving large language diffusion models (e.g., LLaDa(Nie et al., 2025), Dream(Ye et al., 2025)) essentially unaddressed. Overall, unlike in vision, there is no established method for simple, plug-and-play control in text diffusion.

The need for robust controllability is further amplified by the architecture of modern DLMs. State-of-the-art systems adopt *blockwise* diffusion, where the sequence is iteratively refined in contiguous

spans rather than single tokens (Arriola et al., 2025; Nie et al., 2025). Blockwise generation provides efficiency and scalability but also introduces instability: gradient volatility and coverage fluctuate sharply at block boundaries, so a fixed guidance schedule can cause overshoot (semantic drift, spillover of violations) in some blocks or stalling (missed constraints) in others. Existing schedule-based heuristics cannot adapt to such variability, and thus fall short of delivering stable constraint satisfaction.

This paper addresses the central question of *how to choose guidance strength online* so that constraint satisfaction improves predictably while the generation remains fluent and stable under a fixed budget. We introduce **BALE** (Budget-Aware Logit Editing), a plug-and-play controller that replaces heuristic schedules with a principled, online rule. At each step, BALE fuses gradients from multiple lexical and semantic constraints and selects the guidance strength $\lambda_t$ by combining (i) a residual-targeted proposal with two safeguards: (ii) an EMA-based Lipschitz cap that shrinks under volatility spikes and (iii) a KL-aware allowance cap derived from the block's remaining budget. Beyond per-step control, BALE allocates a fixed sequence-level budget across blocks in proportion to anticipated residual "debt," with optional reallocation at block boundaries, promoting smooth progress and budget compliance without manual schedules.

Empirically, we evaluate BALE on LLaDa-8B(Nie et al., 2025) with lexical (must-have, forbidden), semantic (sentiment, formality), and combined constraints. Across all settings, BALE improves constraint satisfaction over prompt-only guidance and schedule-based baselines, often at lower perplexity and with higher human preference. The ablation results further clarify that stability caps and budget-aware rules are not redundant safeguards but complementary: each prevents distinct failure modes, and removing them reintroduces instability.

**Contributions.** (1) We highlight the gap between the relatively mature controllability techniques in vision diffusion and the comparatively limited exploration in text diffusion, where plug-and-play approaches remain sparse. (2) We formulate budget-aware logit guidance for DLMs, combining a residual-target rule and complementary stability caps. (3) We introduce block-level budget allocation with optional dynamic reallocation to adapt to heterogeneous blocks. (4) We provide a lightweight implementation that supports lexical and semantic constraints at inference time without retraining. (5) We present empirical results across multiple tasks and backbones, showing that our online, schedule-free guidance improves controllability and stability over heuristic baselines.

## 2 PRELIMINARIES

### 2.1 FROM LANGUAGE MODELING TO BLOCKED DISCRETE DIFFUSION

Let $\mathcal{X}$ denote a vocabulary of size $V$, and let $x_t \in \mathcal{X}^n$ be a length–$n$ sequence at diffusion step $t$. The *forward noising process* independently replaces each token $x_{t-1}^{(i)}$ with a special mask token [MASK] $\in \mathcal{X}$ with probability $\alpha_t$:

$$q_{\alpha_t}(x_t \mid x_{t-1}) = \prod_{i=1}^{n} \left[ (1 - \alpha_t)\, \delta(x_t^{(i)} = x_{t-1}^{(i)}) + \alpha_t\, \delta(x_t^{(i)} = [\text{MASK}]) \right]. \tag{1}$$

The learned *reverse denoising process* parameterized by $\theta$ predicts temperature–scaled logits $\ell_\theta$ and defines

$$p_\theta(x_{t-1} \mid x_t) \propto \exp\big(\ell_\theta(x_{t-1} \mid x_t)/\tau_t\big), \qquad \tau_t \searrow 0. \tag{2}$$

Blocked diffusion (Nie et al., 2025; Arriola et al., 2025) partitions the sequence into contiguous blocks indexed by $b = 1{:}B_{\text{blocks}}$, each with $T_b$ inner steps. Within block $b$, a remasking/transfer schedule specifies the coverage $r_t \in (0, 1]$, i.e. the fraction of positions updated at inner step $t$.

### 2.2 LOGIT–GUIDED CONTROL IN DISCRETE DIFFUSION

Given the blocked discrete diffusion setup in §2.1, we briefly summarize the generic logit–guided control used throughout this work. At inner step $t$, the base model produces logits $z_t \in \mathbb{R}^{M \times V}$ over the vocabulary and a distribution $p_t = \text{softmax}(z_t)$.

We assume a collection of differentiable control objectives $\{k^{(k)}\}_{k=1}^K$ defined on the logits. Each objective has a non–negative residual

$$\rho_t^{(k)} \;=\; k^{(k)}(\boldsymbol{z}_t) \;\geq 0,$$

which measures the current violation of the constraint, and a gradient $\boldsymbol{g}_t^{(k)} \;=\; \nabla_{\boldsymbol{z}_t} k^{(k)}(\boldsymbol{z}_t)$ with respect to the logits. These objectives can represent, for example, classifier scores, lexical constraint losses, or reward model outputs. To obtain a single update direction, the individual gradients are combined into a fused direction by the fusion rule in §3.3.

A generic logit edit with scalar step size $\lambda_t \geq 0$ takes the form

$$\tilde{\boldsymbol{z}}_t \;=\; \boldsymbol{z}_t - \lambda_t\,\boldsymbol{g}_t, \qquad \boldsymbol{q}_t = \mathrm{softmax}(\tilde{\boldsymbol{z}}_t), \tag{3}$$

applied at the updated positions in block $b$. For small $\lambda_t$, the resulting change between $\boldsymbol{p}_t$ and $\boldsymbol{q}_t$ can be quantified locally via the KL divergence $D_{\mathrm{KL}}(\boldsymbol{p}_t\|\boldsymbol{q}_t)$, providing a notion of per–step edit cost.

### 2.3 LOCAL KL CONTROL AND A COVERAGE–INVARIANT SURROGATE

For per–step accounting we use an information–theoretic control. Let $\boldsymbol{p} = \mathrm{softmax}(\boldsymbol{z})$ and $\boldsymbol{q} = \mathrm{softmax}(\boldsymbol{z} - \lambda\boldsymbol{g})$. Denote the softmax Fisher by $\boldsymbol{F}(\boldsymbol{z}) = (\boldsymbol{p}) - \boldsymbol{p}\boldsymbol{p}^\top$. A second–order expansion around $\lambda = 0$ gives

$$D_{\mathrm{KL}}(\boldsymbol{p}\|\boldsymbol{q}) \;=\; \tfrac{\lambda^2}{2}\,\boldsymbol{g}^\top \boldsymbol{F}(\boldsymbol{z})\,\boldsymbol{g} \;+\; O(\lambda^3) \;\leq\; \tfrac{\lambda^2}{8}\,\|\boldsymbol{g}\|_2^2 \;+\; O(\lambda^3), \tag{4}$$

since $\|\boldsymbol{F}(\boldsymbol{z})\|_2 \leq \tfrac{1}{4}$. To make this scale–free across vocabularies and compatible with masked updates, We track the mean squared gradient $g_{2,t} = \frac{1}{V}\sum_{i=1}^V (g_t^i)^2 = \frac{\|\boldsymbol{g}_t\|_2^2}{V}$, which accounts for gradient volatility. A formal derivation of how partial coverage ($r_t$) affects the aggregate KL cost is provided in Appendix G. This motivates a coverage–normalized budget surrogate

$$b_t \;:=\; \frac{c_0\,\lambda_t^2\,g_{2,t}}{r_t}, \qquad c_0 = \tfrac{1}{4}, \tag{5}$$

with explicit *caps* on $\lambda_t$ that depend on $g_{2,t}$ (see §3.6). Equation 4 upper–bounds the true local KL by a constant multiple of $\lambda_t^2 g_{2,t}$, while equation 5 yields coverage–invariant accounting.

### 2.4 WHY EMPIRICAL $\lambda_t$ IS BRITTLE IN BLOCK DIFFUSION

Prior diffusion–LM guidance typically relies on fixed or linear schedules, temperature–like scalings, or pre–tuned constants for $\lambda_t$ (Li et al., 2022). These heuristics struggle to adapt robustly to three key nonstationarities: (i) volatility spikes ($g_{2,t}$) at block starts or when multiple constraints momentarily align, which can cause overshoot or instability (Analysis are in Appendix H); (ii) varying coverage $r_t$ from remasking/transfer policies, which alters the effective information budget per step; (iii) varying residual magnitudes $\rho_t^{(k)}$ across blocks, so a globally tuned $\lambda_t$ may stall on hard blocks or over–edit easy ones. Because each block offers only $T_b$ inner steps, poor step–size choices are difficult to amortize over time. These observations motivate a *budget–aware*, Lipschitz–capped, *residual–targeted* rule that adapts $\lambda_t$ online, paired with blockwise budget allocation; we develop this controller in §3 and analyze its properties in §5.

## 3 BUDGET-AWARE LOGIT EDITING (BALE): METHODOLOGY

### 3.1 SETTING

We consider masked/iterative *discrete* text diffusion as in §2: the forward process *noises* a sequence by replacing tokens with a special mask token [MASK] at rate $\alpha_t$, and a learned reverse model *denoises* by producing logits. Let $\mathcal{X}$ be a vocabulary of size $V$, and let $x_t \in \mathcal{X}^n$ denote the length–$n$ sequence at inner step $t$.

**Blocks and inner steps.** Generation proceeds in contiguous blocks indexed by $b = 1{:}B_{\mathrm{blocks}}$ (Nie et al., 2025; Arriola et al., 2025). Each block $b$ has $T_b$ inner denoising steps; we denote by $S_b^{\mathrm{left}}$ the number of steps remaining within the active block. A remasking/transfer schedule specifies the *coverage* $r_t \in (0, 1]$, i.e., the fraction of positions updated at step $t$.

**Budgets and per-step edits.** BALE performs a logit edit (equation 3) with guidance strength $\lambda_t$ chosen online. A *global* per-sequence budget $B$ serves as an explicit upper bound on the *total KL drift* allowed during generation. This budget is partitioned into *block-wise* allocations $\{K_b\}_{b=1}^{B_{\text{blocks}}}$ satisfying $\sum_b K_b = B$ (initialized by anticipated debt and optionally reallocated; §3.2). Within each block, BALE treats the budget as a "credit account," tracking cumulative spending via a KL-aware surrogate. Compliance is enforced by (i) an EMA-based Lipschitz cap, (ii) a per-step allowance proportional to the block's remaining budget and steps, and (iii) a residual-targeted proposal for $\lambda_t$ (§3.4–3.6). In this way, the budget guarantees that edits cannot overshoot beyond a fixed credit limit, ensuring stability without hand-tuned schedules.

## 3.2 BLOCK-WISE BUDGET ALLOCATION AND DYNAMIC REALLOCATION

**Initial allocation from anticipated debt.** Given $B$ and blocks $b = 1{:}B_{\text{blocks}}$, we estimate per-block *anticipated debt* from the initial model state. For each block $b$ (prompt offset $s_b$ to $e_b$), we evaluate constraint components on the corresponding logits slice and obtain nonnegative scalars $\{d_{b,k}\}$ (residual magnitudes). We aggregate with task weights to get

$$D_b^{(0)} = \sum_k w_k^{\text{task}} d_{b,k}.$$

We *smooth across neighboring blocks* to avoid brittle spikes:

$$\tilde{D}_b = \tfrac{1}{4} D_{b-1}^{(0)} + \tfrac{1}{2} D_b^{(0)} + \tfrac{1}{4} D_{b+1}^{(0)}$$

and *modulate by local entropy* to invest more budget where the model is uncertain:

$$H_b = \mathbb{E}_{t \in [s_b, e_b]}\big[-\boldsymbol{p}_t \log \boldsymbol{p}_t\big], \quad m_b = 0.5 + 0.5 \cdot \frac{H_b}{\log V} \in [0.5, 1], \quad \widehat{D}_b = m_b \tilde{D}_b.$$

Finally, allocate with a small floor $\varepsilon_B$:

$$K_b = B \cdot \frac{\max\{\widehat{D}_b, \varepsilon_B\}}{\sum_j \max\{\widehat{D}_j, \varepsilon_B\}}. \tag{6}$$

**Dynamic reallocation over remaining blocks.** At the start of block $b^\star$, compute remaining budget $B_{\text{rem}} = B - \sum_{j < b^\star} \sum_{t \in j} b_t^{\text{KL}}$. Re-estimate shares $\{\pi_b\}_{b \geq b^\star}$ over the remaining blocks using the *current* logits (same recipe as above, but restricted to $b \geq b^\star$), and set

$$K_b^{\text{new}} = B_{\text{rem}} \cdot \pi_b, \qquad b \geq b^\star. \tag{7}$$

When reproducibility is critical, we freeze allocation (skip equation 7) and rely on the per-step cap $\lambda_{\text{capB}}$ to respect the remaining $K_b$.

**Per-step allowance and accounting.** Within an active block, define $K_b^{\text{remain}}$ as unspent block budget. The allowance $a_t$ in equation 15 uses $(S_b^{\text{left}} + 1)^\rho$ to amortize the remainder and *automatically tighten* the cap as steps dwindle.

## 3.3 RESIDUAL-WEIGHTED MULTI-CONSTRAINT FUSION

We compute nonnegative weights $\tilde{w}_t^{(k)}$ from residual magnitudes with optional policy hooks ("must-first" amplification, forbiddance caps):

$$\tilde{w}_t^{(k)} \propto \max\{\rho_t^{(k)}, 0\} \cdot \big(\kappa_{\text{must}}\big)^{\mathbf{1}_{\text{must}}(k)} \cdot \min\{1, \kappa_{\text{forbid}}\}^{\mathbf{1}_{\text{forbid}}(k)}, \qquad w_t^{(k)} = \frac{\tilde{w}_t^{(k)}}{\sum_j \tilde{w}_t^{(j)}}.$$

We then fuse directions by a weighted sum to mitigate interference:

$$\boldsymbol{g}_t = \sum_{k=1}^K w_t^{(k)} \boldsymbol{g}_t^{(k)}, \tag{8}$$

followed by RMS normalization $\boldsymbol{g}_t \leftarrow \boldsymbol{g}_t / ((\boldsymbol{g}_t) + \varepsilon)$ for scale stability. The logit edit is $\tilde{\boldsymbol{z}}_t = \boldsymbol{z}_t - \lambda_t \boldsymbol{g}_t$ with $\tilde{\boldsymbol{p}}_t = \text{softmax}(\tilde{\boldsymbol{z}}_t)$.

### 3.4 ONLINE LIPSCHITZ PROXY AND STABILITY CAP

We estimate a local Lipschitz proxy from the pre-normalized control gradient $\tilde{\boldsymbol{g}}_t$. We maintain an EMA of its RMS magnitude:

$$\widehat{L}_{t+1} = \beta\,\widehat{L}_t + (1-\beta)\left((\tilde{\boldsymbol{g}}_t) + \varepsilon_{\mathrm{ema}}\right), \qquad \beta \in (0,1), \tag{9}$$

and impose a stability cap

$$\lambda_{\mathrm{cap}} = \frac{\gamma}{\widehat{L}_t}. \tag{10}$$

In practice we also clamp $\widehat{L}_t \leftarrow \max\{\widehat{L}_t, \varepsilon_{\min}\}$ to avoid division by very small values. A formal argument shows that this cap bounds the single–step amplification; see Appendix B.

### 3.5 KL-AWARE BUDGET ACCOUNTING AND DUAL UPDATE

From the local KL expansion $D_{\mathrm{KL}}(\boldsymbol{p}_t\|\mathrm{softmax}(\boldsymbol{z}_t - \lambda_t\boldsymbol{g}_t)) \lesssim \frac{\lambda_t^2}{8}\|\boldsymbol{g}_t\|_2^2$, we align the *per-step surrogate cost* with the gradient volatility and coverage:

$$b_t^{\mathrm{KL}} = \frac{1}{4}\frac{\lambda_t^2\,g_{2,t}}{r_t}. \tag{11}$$

Within block $b$, let the *nominal per-step allowance* be $c_b = K_b/T_b$. We maintain a nonnegative dual variable via projected ascent:

$$\mu_{t+1} = \left[\mu_t + \alpha\,(b_t^{\mathrm{KL}} - c_b)\right]_+, \tag{12}$$

so overspending increases $\mu_t$, which we interpret as a dual signal of budget pressure. It does not affect step size directly, but we analyze its transient activations in §5.1. We also accumulate global usage $B_{\mathrm{used}} \leftarrow B_{\mathrm{used}} + b_t^{\mathrm{KL}}$.

### 3.6 TWO CAPS AND TARGET: SELECTING $\lambda_t$

We combine a target $\lambda$ with two caps:

$$\textbf{Target:}\quad \Delta_t^\star = c_\Delta \cdot \frac{\sum_k \rho_t^{(k)}}{\max\{1, S_b^{\mathrm{left}}\}}, \qquad \lambda_{\mathrm{target}} = \begin{cases} 0, & g_{2,t} \leq \varepsilon, \\ \Delta_t^\star / \sqrt{g_{2,t} + \varepsilon}, & \text{otherwise,} \end{cases} \tag{13}$$

$$\tag{14}$$

The *stability cap* is equation 10. In addition, we impose a *KL-aware budget cap* from a per-step allowance $a_t$:

$$a_t = \frac{K_b^{\mathrm{remain}}}{(S_b^{\mathrm{left}} + 1)^\rho}, \qquad \lambda_{\mathrm{capB}} = \sqrt{\frac{4\,a_t\,r_t}{g_{2,t} + \varepsilon}}, \tag{15}$$

where $\rho \in [0.5, 1.5]$ controls gentle front-loading ($\rho<1$) vs. conservative late spending ($\rho>1$). The final rule is

$$\lambda_t = \min\Big\{\lambda_{\mathrm{cap}},\, \lambda_{\mathrm{capB}},\, \max\big(\lambda_{\min}, \lambda_{target}\big)\Big\}. \tag{16}$$

### 3.7 STEPWISE AND BLOCKWISE BALE ALGORITHMS

BALE consists of a per–inner-step controller and a lightweight across-block driver (Algorithms BALE–STEP and BALE–DECODE). The step controller: (i) computes residuals/gradients for all constraints and *fuses* them with residual weights and policy hooks (must-first, forbid-cap; §3.3); (ii) forms two caps—a stability cap from an EMA Lipschitz proxy (§3.4) and a KL–allowance cap from the block's remaining budget/steps (§3.5); (iii) sets a residual-mass *target* step and selects $\lambda_t$; (iv) edits the logits and charges the budget with a KL surrogate. This yields stable progress and tight per-block budget tracking with about one extra backward pass per step.

The block driver initializes per-block budgets via anticipated debt (§3.2), iterates blocks left→right while resetting the dual/EMA at boundaries, and (optionally) reallocates the unspent global budget to future blocks (§3.2). The same interface applies to AR decoding by using block size 1; KV caching is unaffected since only logits are edited. BALE is plug-and-play: it requires only the logits stream and differentiable constraint heads, and leaves the base sampler unchanged. Implementation details, complexity analysis, and default hyperparameters are provided in Appendix C.

**Algorithm 1** BALE–STEP (within a block)

**Input** : $z_t, \{\rho_k\}, \{g_k\}, K_b^{\text{rem}}, \mu, \widehat{L}$
**Output:** $x_{t+1}, K_b^{\text{rem}}, \mu, \widehat{L}$

1 $g \leftarrow \text{fuse\_and\_normalize}(\{\rho_k\}, \{g_k\})$
  $\lambda_{\text{cap}}, \lambda_{\text{capB}} \leftarrow \text{caps}$   // Eq. (10), (15)
2 $\lambda_{\text{target}} \leftarrow \text{target}$            // Eq. (13)
3 $\lambda_t \leftarrow \min\left(\lambda_{\text{cap}}, \lambda_{\text{capB}}, \lambda_{\text{target}}\right)$
4 $\tilde{z}_t \leftarrow z_t - \lambda_t g$
5 $x_{t+1} \leftarrow \text{sample}(\text{softmax}(\tilde{z}_t))$
6 $b_t^{\text{KL}} \leftarrow \text{KL}(\text{softmax}(\tilde{z}_t), \text{softmax}(z_t))$
7 $K_b^{\text{rem}} \leftarrow K_b^{\text{rem}} - b_t^{\text{KL}}$ , Update $\mu, \widehat{L}$
  // Eq. (9), (12)
8 **return** $x_{t+1}, K_b^{\text{rem}}, \mu, \widehat{L}$

**Algorithm 2** BALE–DECODE (across blocks)

**Input** : #blocks $B$, global budget $B_{\text{tot}}$ (or per-block $\{K_b\}$), base decoder producing logits $\{z_t\}$
9 Initialize per-block $K_b$, set $K_b^{\text{rem}} \leftarrow K_b$
10 **for** $b = 1$ **to** $B$ **do**
11   Reset $\mu \leftarrow 0, \widehat{L} \leftarrow \epsilon$; steps-left $S \leftarrow T_b$
    **while** *block b active* **do**
12     $(x_{t+1}, K_b^{\text{rem}}, \mu, \widehat{L}) \leftarrow \text{BALESTEP}(\ldots)$
      Advance decoder to next $z_t$;
      $S \leftarrow S - 1$
13   Optional reallocate unused global budget to future blocks
**Output:** final sequence $x$

## 4 Experiments and Main Results

### 4.1 Setup at a Glance

We evaluate three control types: *semantic* control—targeting negative sentiment and informal style—on WritingPrompts (Fan et al., 2018), *lexical* control—keyword injection—on CommonGen-style concept sets (Lin et al., 2019) and combined control with multiple constraints. All systems share a single diffusion backbone (LLaDa-8B-Instruct (Nie et al., 2025)) and include: (i) vanilla prompt-only guidance (PO), (ii) three schedule-based baselines with constant, linear, or cosine decay of the guidance scale, and (iii) our budgeted controller (**BALE**), which interprets the scale as a sequence-level budget and allocates it adaptively over steps. Main results are reported on three random seeds, while analysis settings are evaluated on a shared random seed. Effectiveness is measured by *control accuracy*, defined as ensemble-classifier agreement with the target constraint, and fluency by *relative perplexity* (R_PPL), which augments perplexity with diversity penalties. For lexical control, we report *coverage accuracy* (all required keywords realized) and *positional accuracy* (keywords placed in grammatically valid contexts). Full details are deferred to Appendix D.

### 4.2 Semantic Control: Sentiment and Formality

Table 1 reports results on WritingPrompts. For sentiment control, BALE+PO achieves the highest accuracy (**0.97**) while also reducing perplexity (**9.09** vs. 10.04–12.37). For formality, BALE+PO improves accuracy to **0.93** with only a modest fluency cost (8.95 vs. 8.22–9.91). These results indicate that constant step sizes tend to over-steer (high accuracy but poor fluency) and decay schedules trade accuracy for fluency, whereas BALE provides stronger semantic steering while keeping fluency competitive.

Table 1: WritingPrompts semantic control. Each cell shows Acc. ($\uparrow$) / R_PPL ($\downarrow$).

| Method | Negative | Informal |
|---|---|---|
| PO | 0.80 / 5.65 | 0.70 / 6.54 |
| Const. $\lambda$ + PO | 0.95 / 12.37 | 0.91 / 9.91 |
| Linear decay + PO | 0.94 / 10.34 | 0.89 / 9.58 |
| Cosine decay + PO | 0.95 / 10.04 | 0.90 / 9.48 |
| **BALE + PO** | **0.97 / 9.09** | **0.93 / 8.95** |

### 4.3 Lexical Control: Keyword Injection

Table 2: CommonGen keyword injection. Acc. ($\uparrow$), POS. ($\uparrow$, positional accuracy), R_PPL($\downarrow$)

| Method | Acc.% | POS.% | R_PPL |
|---|---|---|---|
| PO | 94.44 | 72.00 | 18.02 |
| Const. $\lambda$ + PO | 93.44 | 67.56 | **22.43** |
| Cosine decay + PO | 94.67 | 73.00 | 22.68 |
| **BALE + PO** | **96.67** | **73.11** | 24.64 |

On CommonGen keyword injection, **BALE**+PO attains the strongest coverage (96.7%), improving over PO and constant-$\lambda$. It also yields the highest positional accuracy, indicating that keywords are naturally integrated into grammatical roles rather than appended superficially. While BALE increases R_PPL, the fluency trade-off is modest relative to the accuracy gains.

### 4.4 COMBINED CONTROL

[1] We evaluate combined control under two settings: lexical $\oplus$ semantic (keywords + negative sentiment) and lexical $\oplus$ lexical (keywords + forbidden words).

BALE attains the highest joint success in both cases, improving over PO and all schedule baselines while keeping R_PPL competitive. On key$\oplus$neg, BALE reaches 0.875 joint success with lower R_PPL than schedule baselines, while on key$\oplus$forb it improves to 0.695 from 0.650–0.668 under baselines. These results demonstrate that budget-aware fusion maintains robustness even when heterogeneous constraints interact.

Table 3: Combined control results. Each cell shows **Joint success** ($\uparrow$) / **R_PPL** ($\downarrow$).

| Method | key$\oplus$neg[†] | key$\oplus$forb[‡] |
|---|---|---|
| PO | 0.788 / 15.87 | 0.668 / 12.91 |
| Const. $\lambda$+PO | 0.685 / 19.12 | 0.650 / 13.40 |
| Linear decay + PO | 0.690 / 18.94 | 0.655 / 13.44 |
| Cosine decay + PO | 0.735 / 19.01 | 0.660 / 13.64 |
| **BALE + PO** | **0.875 / 18.76** | **0.695 / 13.20** |

### 4.5 HUMAN PREFERENCE JUDGMENTS

To validate BALE's effectiveness, we conducted blind A/B preference tests against the strongest schedule-based baseline. Human evaluators are tasked with selecting the better generation on two combined control tasks-**keyword$\oplus$negative** and **keyword$\oplus$forbidden**. Detailed information about human evaluation is in the Appendix E.

As summarized in Figure 1, the human assessors decisively preferred BALE in both tasks. In the keyword$\oplus$sentiment task, BALE showed its ability to satisfy multiple constraints without sacrificing quality. Additionally, BALE achieved improvements in mitigating the degenerate outputs often produced by schedule-based methods. Ultimately, these findings show that BALE's principled approach not only improves automatic metrics but also yields outputs that are judged to be more correct and fluent.

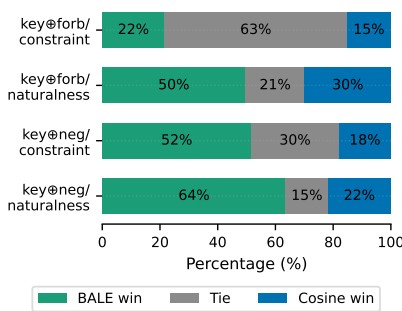

Figure 1: Human preference

### 5 ANALYSIS AND ABLATIONS

#### 5.1 $\lambda$ DECISION BREAKDOWN AND BLOCKWISE DYNAMICS

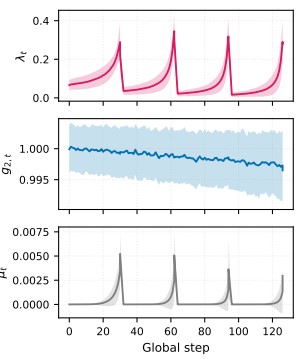

Figure 2: Dynamics of $\lambda_t$, $g_{2,t}$, and $\mu_t$ across steps.

We study how BALE adapts $\lambda_t$ across blocks and steps, contrasting it with fixed schedule baselines. Figure 2 shows the dynamics of $\lambda_t$, $g_{2,t}$, and $\mu_t$ across steps. Unlike hand-designed schedules that monotonically decay regardless of context, BALE adjusts $\lambda_t$ online: it increases transiently when residuals rise and decays as constraints are gradually satisfied. This adaptivity is useful because residual magnitudes and block coverages vary across steps, which fixed schedules cannot anticipate. In our experiments, fixed schedules sometimes produced large early updates or insufficient late corrections, whereas BALE followed smoother decay while remaining responsive to residual signals. Meanwhile, $g_{2,t}$ stays stable, and the dual variable $\mu_t$ activates briefly, coinciding with moments of overspending. These observations suggest that budget-aware control contributes to stability against local spikes without hindering late-stage fluency.

Blockwise effects are further illustrated in Figure 3. The left panel shows the breakdown of $\lambda_t$ decisions across blocks. Most updates follow the `target` rule, while transient activations of the EMA stability cap (`cap`) and the KL-aware budget cap (`capB`) appear

---

[1][†] Joint success = [all keywords covered] $\wedge$ [sentiment=negative]. [‡] Joint success = [all keywords covered] $\wedge$ [no forbidden hit].

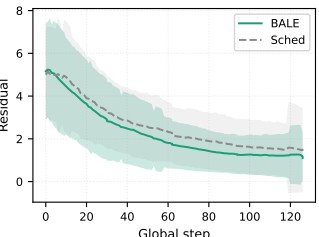
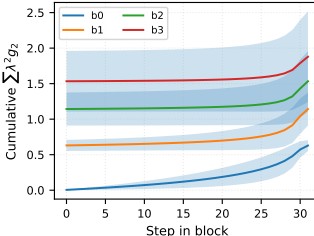

Figure 3: Lambda decision breakdown on blocks(left), residual decay (middle) and cumulative KL surrogate (right). BALE reduces residuals more smoothly and suppresses early KL spikes compared to schedule-only baselines.

<table>
<tr><td colspan="3">Table 4: Ablation Study</td></tr>
</table>

| Method | Succ. | R_PPL |
|--------|-------|-------|
| BALE - cap | 0.845 | 18.805 |
| BALE - capB | 0.860 | 18.787 |
| BALE - both | 0.845 | 18.953 |
| **BALE** | **0.875** | **18.756** |

Table 5: Backbone Gen.

| Method | Succ. | R_PPL |
|--------|-------|-------|
| PO | 0.540 | 18.337 |
| Const.+PO | 0.585 | 18.775 |
| Cosine+PO | 0.618 | 17.433 |
| **BALE+PO** | **0.635** | **17.097** |

Table 6: vs. AR Models

| Method | Succ. | R_PPL |
|--------|-------|-------|
| AR+PO | 0.685 | 19.528 |
| AR+DExperts | 0.705 | 29.360 |
| AR+DeAL | 0.730 | 20.790 |
| **BALE+PO** | **0.875** | **18.756** |

near block boundaries. These activations often coincide with reductions in abrupt early updates, which are not mitigated by schedule-only methods. The middle panel shows that BALE reduces residuals more smoothly than heuristic schedules, indicating more efficient constraint satisfaction under our tasks. Finally, the right panel plots cumulative KL within blocks:We can see that BALE distributes budget usage evenly. Together, these results indicate that online budget-aware control helps suppress instabilities observed with schedule-only baselines, while maintaining controllability throughout the block.

## 5.2 ABLATION STUDY

We begin by isolating the contribution of BALE's safeguards on the keyword⊕negative combined task. As shown in Table 4, removing either the EMA/volatility cap or the budget-aware cap reduces joint success ($0.875 \rightarrow 0.860/0.845$) and slightly worsens fluency (R_PPL $18.756 \rightarrow 18.787/18.805$). Dropping both caps further exacerbates the effect ($0.845 / 18.953$). These findings confirm that (i) the EMA-based cap suppresses step-size explosions at block starts, and (ii) the budget-aware cap prevents overspending spikes in later steps.

## 5.3 BACKBONE GENERALIZATION

We next examine BALE on a different diffusion backbone, Dream-7B(Ye et al., 2025), while using the same prompt-only head (PO). Table 5 shows that BALE improves joint success from 0.618 under schedule-only control (Const./Cosine+PO) to 0.635, while reducing the relative perplexity of schedule-only baselines ($17.433 \rightarrow 17.097$). This demonstrates that BALE's adaptive step-size selection generalizes beyond LLaDa, yielding smoother residual decay and more stable behavior across architectures.

## 5.4 COMPARISON WITH AUTOREGRESSIVE MODELS

Finally, we compare BALE with autoregressive (AR) control baselines using the Mistral-7B-v0.3 backbone(Jiang et al., 2023). As reported in Table 6, AR+PO achieves only 0.685 success, while advanced plug and play controllers such as DExperts(Liu et al., 2021) and DeAL(Huang et al., 2024) either sacrifice fluency or plateau in success. In contrast, BALE+PO applied on diffusion baselines achieves the best overall trade-off. Importantly, BALE requires only one backward pass per step for active constraints and an $\mathcal{O}(1)$ dual/cap update, keeping throughput comparable to schedule-only diffusion.

## 6 RELATED WORK AND LIMITATIONS

**Diffusion-LM.** To overcome the sequential limitations of autoregressive (AR) decoders (Gu et al., 2017; Stern et al., 2019), diffusion models from vision tasks (Sohl-Dickstein et al., 2015; Ho et al., 2020; Song et al., 2020) have emerged as a new non-autoregressive (NAR) paradigm for language models. Foundational methods such as Diffusion-LM (Li et al., 2022) and DiffuSeq (Gong et al., 2022) established the core principle of generating coherent text via iterative denoising by leveraging global context at each step (Savinov et al., 2021; He et al., 2022; Austin et al., 2021). This paradigm has been successfully scaled to large models such as LLaDa (Nie et al., 2025) and Dream (Ye et al., 2025). However, precisely controlling these models to satisfy specific constraints remains a significant challenge, as early guidance techniques often relied on brittle, heuristic schedules that required extensive task-specific tuning (Dhariwal & Nichol, 2021; Balaji et al., 2023).

**AR plug-and-play control.** Autoregressive LMs have long supported decoding-time control through gradient-based editing(Dathathri et al., 2019), expert reweighting(Liu et al., 2021), and discriminator guidance(Krause et al., 2020; Yang & Klein, 2021). More recent work explores alignment-oriented decoding, including DeAL for decoding-time alignment(Huang et al., 2024) or personalized alignment during decoding(Kim et al., 2025). While effective, these methods often depend on heuristics or auxiliary discriminators, and their stability across prompts remains limited. BALE follows the same plug-and-play ethos but introduces explicit KL budgeting and dual updates for robustness, and applies uniformly to both diffusion and AR decoders.

**Control for diffusion models.** Guidance in diffusion has been most widely explored in continuous domains. Classifier guidance (Dhariwal & Nichol, 2021), classifier-free guidance (Ho & Salimans, 2022), and reward-based steering (Zhang & Xu, 2023; Yuan et al., 2023) have become standard tools for plug-and-play control. For text diffusion, early studies have experimented with classifier losses (sentiment, toxicity), bag-of-words cues, and heuristic schedules (Li et al., 2022; Gong et al., 2022; Han et al., 2022), and some methods even re-train models with control objectives (Cardei et al., 2025). More recently, Schiff et al. (2024) proposed guidance for discrete diffusion by training diffusion-time classifiers matched to the underlying corruption process and scaling their gradients with a fixed global scaling factor. BALE is complementary: it assumes an existing differentiable control signal and focuses on *how strongly and when* to apply it via an online, state-aware controller $\lambda_t$ that allocates a KL budget across blocks and steps. This lets BALE wrap off-the-shelf classifiers without diffusion-specific retraining and provide predictable degradation bounds without hand-tuned schedules.

**Limitations.** BALE relies on differentiable surrogate losses for steering, so constraints that are inherently symbolic are not directly supported and must be approximated, potentially introducing mismatch between the intended rule and the optimized signal. The controller adds one backward pass per decoding step, which yields modest runtime overhead in diffusion blocks but may become more noticeable for very small coverage or pure AR decoding. Because step sizes are determined from local residuals and a KL allowance, BALE does not currently model long-range discourse, which can limit global structural control. Finally, both steering and evaluation inherit biases from off-the-shelf classifiers used as control heads; while ensembles and human preference checks help mitigate these effects, removing such biases entirely remains an open challenge.

## 7 CONCLUSION

We introduced **BALE**, a budget-aware controller for constraint-guided logit decoding in blocked text diffusion. BALE combines a dual-based budget tracker, a residual-aware target rule, and an EMA stability cap to steer generation while preserving the stability of the base decoder, without requiring retraining. Empirically, BALE achieves stronger controllability across lexical (coverage, position) and semantic (sentiment, formality) constraints, and improves joint success on combined tasks, often at comparable or better generation quality than offline scheduled $\lambda$ baselines. Human preference evaluation further supports that budgeted steering raises constraint satisfaction without compromising fluency. Future directions include adaptive calibration across prompts, extending to richer structural constraints, and exploring integration with preference optimization methods for multi-attribute control at scale.

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

## A  CONTINUOUS-TIME VIEW: FP INTERPRETATION AND MEAN-SHIFT BOUND

## B  STABILITY UNDER EMA LIPSCHITZ CAP

This appendix establishes a stability guarantee for the EMA-based Lipschitz cap introduced in §3.4. Recall that BALE constrains the guidance strength by $\lambda_t \leq \gamma/\widehat{L}_t$, where $\widehat{L}_t$ is an EMA proxy of the local Lipschitz constant of the (pre-normalized) control vector field. We show that this rule *bounds the single-step amplification* of the logit update, thereby curbing step-induced instability.

**Lemma B.1 (Bounded amplification under EMA cap)** *Let $G_t$ denote one editing step with step size $\lambda_t$ and let $v_t$ be the (pre-normalized) control field with local Lipschitz constant $L_t$, i.e., $\|v_t(\boldsymbol{\ell}_1) - v_t(\boldsymbol{\ell}_2)\| \leq L_t \|\boldsymbol{\ell}_1 - \boldsymbol{\ell}_2\|$. If the stability cap enforces $\lambda_t \leq \gamma/\widehat{L}_t$ and $\widehat{L}_t \geq L_t$, then for all logits $\boldsymbol{\ell}_1, \boldsymbol{\ell}_2$,*

$$\|G_t(\boldsymbol{\ell}_1) - G_t(\boldsymbol{\ell}_2)\| \;\leq\; \big(1 + \lambda_t L_t\big)\,\|\boldsymbol{\ell}_1 - \boldsymbol{\ell}_2\| \;\leq\; (1+\gamma)\,\|\boldsymbol{\ell}_1 - \boldsymbol{\ell}_2\|.$$

*Thus the edit map has single–step amplification bounded by $(1+\gamma)$.*

Write $G_t(\boldsymbol{\ell}) = \boldsymbol{\ell} + \lambda_t\, v_t(\boldsymbol{\ell})$. By the triangle inequality and the Lipschitz property of $v_t$,

$$\|G_t(\boldsymbol{\ell}_1) - G_t(\boldsymbol{\ell}_2)\| \;\leq\; \|\boldsymbol{\ell}_1 - \boldsymbol{\ell}_2\| + \lambda_t \|v_t(\boldsymbol{\ell}_1) - v_t(\boldsymbol{\ell}_2)\| \;\leq\; \big(1 + \lambda_t L_t\big)\,\|\boldsymbol{\ell}_1 - \boldsymbol{\ell}_2\|.$$

Whenever $\widehat{L}_t \geq L_t$ and $\lambda_t \leq \gamma/\widehat{L}_t$, we have $\lambda_t L_t \leq \gamma$, which yields the claim.

**Corollary (Iterated stability).**  If the condition of Lemma B.1 holds for $t = t_0, \ldots, t_1$, then the composed map $G_{t_1} \circ \cdots \circ G_{t_0}$ satisfies

$$\|G_{t_1:t_0}(\boldsymbol{\ell}_1) - G_{t_1:t_0}(\boldsymbol{\ell}_2)\| \;\leq\; \prod_{t=t_0}^{t_1} (1+\gamma_t)\,\|\boldsymbol{\ell}_1 - \boldsymbol{\ell}_2\|,$$

where $\gamma_t := \lambda_t \widehat{L}_t \leq \gamma$. For small $\gamma$, $\log\big(\prod_t (1+\gamma)\big) = O(\gamma\,(t_1 - t_0 + 1))$, indicating controlled growth.

(i) The proxy $\widehat{L}_t$ is computed from the *pre-normalized* gradient RMS (see §3.4), while the applied update uses an RMS-normalized direction; this decouples magnitude tracking from the step direction. (ii) Although $\widehat{L}_t$ is an EMA *proxy* and need not upper-bound $L_t$ at every step, the bound above holds whenever $\widehat{L}_t \geq L_t$. To bias the proxy from above in practice, one may maintain an upper envelope,

$$\widehat{L}_{t+1}^{\uparrow} \;\leftarrow\; \max\Big\{\widehat{L}_{t+1},\, \max_{s \in [t-w,\, t]}(\tilde{\boldsymbol{g}}_s)\Big\},$$

and use $\widehat{L}_t^{\uparrow}$ in the cap. (iii) We also clamp $\widehat{L}_t \leftarrow \max\{\widehat{L}_t, \varepsilon_{\min}\}$ for numerical safety.

In summary, capping $\lambda_t$ by $\gamma/\widehat{L}_t$ yields a *bounded-amplification* guarantee per step, which complements the empirical evidence in §5.1 that the cap suppresses instability while preserving late-stage fluency.

## C  COMPLEXITY AND DEFAULT HYPERPARAMETERS

**Per-step complexity.**  BALE adds one backward pass through the *active* control heads per inner step and $O(KV)$–$O(V)$ vector ops for fuse/normalize, where $K$ is the number of active constraints and $V$ the vocab size. EMA/dual/cap updates are $O(1)$. The cache and sampler are unchanged.

**Model and precision.**  Unless otherwise noted, we use `GSAI-ML/LLaDA-8B-Instruct` with `bfloat16` weights and CUDA environments. Tokenizer padding is set to EOS if missing, and `mask_token_id` is taken from the tokenizer or model config.

**Decoding defaults.** For all runs in the main experiments we use:

$$\texttt{steps} = 128, \quad \texttt{gen\_length} = 128, \quad \texttt{block\_length} = 32,$$

temperature $= 0.2$, CFG scale $= 0.0$ (i.e., disabled), and the remasking policy `low_confidence`. These settings are held fixed across BALE and schedule baselines.

**Controller (BALE) configuration.** When `mode=bale`, the budget controller is instantiated as:

$$\texttt{total\_budget} = 3, \ \ \alpha = 0.1, \ \ \gamma_{\text{scale}} = 2.0,$$

$$\texttt{delta\_target\_coef} = 0.4, \ \ \rho\_\text{scale} = 0.25,$$

Here, $\alpha$ is the dual ascent step (§. 3.5), $\gamma_{\text{scale}}$ maps to the EMA Lipschitz cap factor (§. 3.4), `delta_target_coef` corresponds to the residual-target coefficient $c_\Delta$ (§. 3.6), $\rho\_\text{scale}$ shapes the per-step allowance (equation 15).

## D EXPERIMENTAL SETUP

### D.1 MODELS AND DECODING REGIMES

We evaluate all methods on blocked diffusion decoders where each block contains $T_b$ refinement steps with confidence-based remasking. Unless specified otherwise, block length is fixed at 32 and the remasking policy is `low_confidence`. BALE modifies only the logits, leaving the underlying sampler and KV cache unchanged.

### D.2 TASKS AND DATASETS

**Lexical control (CommonGen).** We sample 400 concept sets from the official validation split. Each set contains 3–5 concepts, all of which must be realized in the output.

**Sentiment control (WritingPrompts).** We select 600 prompts and target *negative* sentiment, extending each prompt into a 3–5 sentence continuation under guidance.

**Formality control (WritingPrompts).** We use 600 prompts and target the *informal* style, analogous to the sentiment setting.

**Combined control (CommonGen).** We sample 400 concept sets, each with 3–5 required concepts plus either a target negative sentiment or forbidden token set.

### D.3 CONSTRAINT HEADS

**Keyword inclusion.** Must-have and forbidden vocabularies are mapped to subword token IDs; multi-token phrases are supported through sequence matching.

**Sentiment head.** We train a lightweight MLP classifier (hidden size 1024) on pooled LM embeddings, uniformly labeled across {negative, neutral, positive}.

**Formality head.** A parallel classifier (hidden size 1024) is trained for three-way labels {informal, casual, formal}.

### D.4 BASELINES

We compare against: (1) **Promp-Only guidance (PO)** without no additional control. (2) **Constant guidance + PO**, with a fixed $\lambda$. (3) **Linear decay + PO** and **Cosine decay + PO**, where $\lambda_t$ follows a pre-specified schedule. (4) **BALE + PO** (ours), replacing fixed schedules with budget-aware allocation.

All systems use the same backbone (LLaDa-8B-Instruct). Decoding settings are identical across methods: temperature 0.2, CFG scale 0.0, block length 32, and generation length 128.

## D.5 Evaluation Metrics

**Task-level metrics.** For lexical control we report: (i) *Coverage accuracy*: proportion of outputs containing all required concepts, (ii) *Positional accuracy*: fraction where concepts appear in grammatically correct positions, and (iii) *Forbidden violation rate*: proportion of outputs containing forbidden tokens.

For sentiment we use accuracy of an ensemble of external classifiers: `distilbert-base-uncased-finetuned-sst-2-english`, `textattack/roberta-base-SST-2`, and `cardiffnlp/twitter-roberta-base-sentiment-latest`.

For formality we use three classifiers: `LenDigLearn/formality-classifier-mdeberta-v3-base`, `MoritzLaurer/deberta-v3-large-zeroshot-v2.0`, and `ggallipoli/formality_classifier_gyafc_family`.

**Fluency metric.** We compute relative perplexity (R_PPL) using a frozen LLaMA-2-13B scorer:

$$\text{R\_PPL} = \text{PPL} \times \big(1 + \alpha(1 - \text{Dist2}) + \beta(1 - \text{Dist3})\big) \times (L/120)^{\delta},$$

with $(\alpha, \beta, \delta) = (1, 2, 0.5)$. Dist-n penalizes repetitions, and the length term normalizes to the 128-token budget.

**Decoding-health metrics.** We track average length, repetition-3 (rep-3), Distinct-2, and report optional human judgments on a 100-sample subset.

**Control metrics.** We measure budget usage ratio $\sum_t b_t / K$, the trajectory of the dual variable $\mu_t$, and per-step KL charges. Wall-clock time and memory overhead are also logged.

# E Human Evaluation

We conducted human evaluations using a custom web interface hosted at `URL_hidden_for_double-blind_review`. The study followed a blind A/B comparison protocol: for each prompt, participants were shown two generations, one from **BALE** and one from the strongest schedule-based baseline, presented in randomized order without system identifiers.

**Participants.** We recruited five participants, all either undergraduate or graduate students with an engineering background. All participants were fluent in English reading comprehension. Each participant provided informed consent and received a small compensation for their time.

**Tasks.** We evaluated two combined-control settings: (i) **keyword⊕negative sentiment** and (ii) **keyword⊕forbidden**. For each task, 100 prompts (or concept sets) were sampled from the evaluation split.

**Procedure.** Each participant evaluated a total of 200 prompt–generation pairs (100 per task). For each pair, two questions were asked:

1. Which output better satisfies the listed constraints?

2. Which output is more natural and fluent?

For each question, evaluators could select (a) Model A's output, (b) Model B's output, or (c) "tie" if both were judged equally good or bad. The order of presentation was randomized to mitigate positional bias. The average evaluation time per participant was 60–70 minutes.

**Measurement.** We report aggregated results in terms of *Win–Tie–Lose (WTL)* counts for BALE against the baseline, computed separately for constraint satisfaction and fluency. A "win" indicates that BALE's output was preferred, "lose" indicates the baseline was preferred, and "tie" indicates no preference. The WTL counts are then converted to preference rates at the task level.

**Statistical Significance.** Beyond raw WTL counts, we applied binomial tests (excluding ties) to assess whether BALE's win rates were significantly above chance. Results show consistent advantages: for **sentiment**, BALE achieved 74.1% win rate on constraint satisfaction ($p < 10^{-19}$) and 74.5% on fluency ($p < 10^{-24}$); for **forbidden**, BALE reached 58.7% ($p = 0.011$) and 62.5% ($p < 10^{-6}$), respectively. Approximate 95% confidence intervals were within $\pm 5\%$. While tie rates were relatively high (e.g., 63% in the forbidden–constraint condition), the results consistently favor BALE.

## F  STOCHASTIC MULTI-SEED EVALUATION

This section reports the full multi-seed evaluation results for main experiments in the paper.

### F.1  SEMANTIC CONTROL (WRITINGPROMPTS) — TABLE 1

Table 7: Semantic control on WritingPrompts (3 seeds). Each cell reports mean $\pm$ std of Acc. ($\uparrow$) / R_PPL ($\downarrow$).

| Method | Negative | Informal |
|---|---|---|
| PO | $0.80_{\pm 0.0278}$ / $5.65_{\pm 0.1508}$ | $0.70_{\pm 0.0229}$ / $6.54_{\pm 0.0707}$ |
| Const. $\lambda$ + PO | $0.95_{\pm 0.0000}$ / $12.37_{\pm 0.0639}$ | $0.91_{\pm 0.0000}$ / $9.91_{\pm 0.0181}$ |
| Linear decay + PO | $0.94_{\pm 0.0058}$ / $10.34_{\pm 0.0980}$ | $0.89_{\pm 0.0029}$ / $9.58_{\pm 0.0677}$ |
| Cosine decay + PO | $0.95_{\pm 0.0029}$ / $10.04_{\pm 0.0163}$ | $0.90_{\pm 0.0000}$ / $9.48_{\pm 0.0166}$ |
| **BALE + PO** | $\mathbf{0.97}_{\pm \mathbf{0.0050}}$ / $\mathbf{9.09}_{\pm \mathbf{0.0476}}$ | $\mathbf{0.93}_{\pm \mathbf{0.0050}}$ / $\mathbf{8.95}_{\pm \mathbf{0.0670}}$ |

### F.2  LEXICAL CONTROL (COMMONGEN) — TABLE 2

Table 8: Keyword control on CommonGen (3 seeds). Each cell reports mean $\pm$ std of Acc.% ($\uparrow$), POS.% ($\uparrow$), and R_PPL ($\downarrow$).

| Method | Acc.% | POS.% | R_PPL |
|---|---|---|---|
| PO | $94.44_{\pm 0.58}$ | $72.00_{\pm 0.76}$ | $18.02_{\pm 0.58}$ |
| Const. $\lambda$ + PO | $93.44_{\pm 0.58}$ | $67.56_{\pm 0.76}$ | $\mathbf{22.43}_{\pm \mathbf{0.38}}$ |
| Linear decay + PO | $93.11_{\pm 0.29}$ | $70.11_{\pm 1.04}$ | $22.82_{\pm 0.65}$ |
| Cosine decay + PO | $94.67_{\pm 0.50}$ | $73.00_{\pm 0.50}$ | $22.64_{\pm 0.42}$ |
| **BALE + PO** | $\mathbf{96.67}_{\pm \mathbf{0.29}}$ | $\mathbf{73.11}_{\pm \mathbf{0.29}}$ | $24.64_{\pm 0.22}$ |

### F.3  COMBINED CONTROL (KEYWORD ⊕ NEGATIVE) AND (KEYWORD ⊕ FORBIDDEN) — TABLE 3

Table 9: Combined control on WritingPrompts + CommonGen (3 seeds). Each cell reports mean $\pm$ std of **Joint success** ($\uparrow$) / **R_PPL** ($\downarrow$).

| Method | key⊕neg[†] | key⊕forb[‡] |
|---|---|---|
| PO | $0.788_{\pm 0.0039}/15.87_{\pm 0.4257}$ | $0.668_{\pm 0.0114}/12.91_{\pm 0.3088}$ |
| Const. $\lambda$ + PO | $0.685_{\pm 0.0072}/19.12_{\pm 0.2214}$ | $0.650_{\pm 0.0151}/13.40_{\pm 0.4707}$ |
| Linear decay + PO | $0.690_{\pm 0.0752}/18.94_{\pm 1.1173}$ | $0.655_{\pm 0.0180}/13.44_{\pm 0.0883}$ |
| Cosine decay + PO | $0.735_{\pm 0.0563}/19.01_{\pm 1.2451}$ | $0.660_{\pm 0.0144}/13.64_{\pm 0.0884}$ |
| **BALE + PO** | $\mathbf{0.875}_{\pm \mathbf{0.0153}}/18.76_{\pm \mathbf{0.2927}}$ | $\mathbf{0.695}_{\pm \mathbf{0.0050}}/13.20_{\pm \mathbf{0.2504}}$ |

Across all main experimental settings, BALE consistently outperforms the strongest schedule-based baselines in controllability metrics. On the combined-control tasks, BALE achieves improvements of about **2.4–2.5 standard deviations** in joint success, and the margins are even larger (up to **6–7**

standard deviations) on semantic control. At the same time, text quality (as measured by R_PPL) remains comparable to or competitive with the best schedule-based configurations, confirming that BALE's gains are statistically robust without sacrificing fluency.

# G  KL SURROGATE AND BUDGET DERIVATIVES

**Derivation from Eq. 4 to Eq. 11.**   We start from the local KL upper bound (Eq. 4):

$$D_{\mathrm{KL}}(\boldsymbol{p}\|\boldsymbol{q}) = \tfrac{\lambda^2}{2}\,\boldsymbol{g}^\top \boldsymbol{F}(\boldsymbol{z})\,\boldsymbol{g} + O(\lambda^3) \leq \tfrac{\lambda^2}{8}\,\|\boldsymbol{g}\|_2^2 + O(\lambda^3), \tag{17}$$

where $\|\boldsymbol{F}(\boldsymbol{z})\|_2 \leq \frac{1}{4}$. For the controller we drop the $O(\lambda^3)$ term and work with

$$D_{\mathrm{KL},t} \leq \frac{\lambda_t^2}{8}\,\|\boldsymbol{g}_t\|_2^2. \tag{18}$$

We then define the mean–squared gradient (MSG)

$$g_{2,t} := \frac{1}{V}\sum_{i=1}^{V}(g_{t,i})^2 = \frac{\|\boldsymbol{g}_t\|_2^2}{V}, \tag{19}$$

so that $\|\boldsymbol{g}_t\|_2^2 = V\,g_{2,t}$. Substituting equation 19 into equation 18 yields

$$D_{\mathrm{KL},t} \leq \frac{\lambda_t^2}{8}\,V\,g_{2,t}. \tag{20}$$

**(a) Why the vocab size $V$ disappears.**   Eq. equation 20 can be written as

$$D_{\mathrm{KL},t} \leq \underbrace{\frac{V}{8}}_{\text{global constant}} \lambda_t^2 g_{2,t}. \tag{21}$$

In our budget controller, we do not act directly on $D_{\mathrm{KL},t}$, but on a surrogate cost $b_t$ whose sum is constrained:

$$\sum_t b_t \leq K_b, \tag{22}$$

for some global budget $K_b$. If we replace $b_t$ by $\kappa b_t$ for any $\kappa > 0$, then we can equivalently rescale the budget and dual step size as

$$K_b \mapsto \kappa K_b, \quad \alpha \mapsto \alpha/\kappa,$$

without changing the controller's behavior. Thus the factor $V/8$ in equation 20 can be absorbed into the global calibration of $(K_b, \alpha)$, and we work in $\lambda_t^2 g_{2,t}$ that are scale–free across vocabularies. In this sense, $V$ does not vanish mathematically, but is absorbed into the global budget.

**(b) Why $1/8$ becomes $1/4$.**   From equation 20, a natural surrogate (ignoring coverage for the moment) is

$$\tilde{b}_t := \lambda_t^2 g_{2,t}, \tag{23}$$

for which

$$D_{\mathrm{KL},t} \leq \left(\frac{V}{8}\right)\tilde{b}_t. \tag{24}$$

More generally, we can introduce a calibration constant $c_0 > 0$ and define

$$b_t^{\mathrm{KL}} := c_0\,\lambda_t^2 g_{2,t}, \tag{25}$$

so that

$$D_{\mathrm{KL},t} \leq \underbrace{\frac{V}{8c_0}}_{:=\kappa}\,b_t^{\mathrm{KL}}. \tag{26}$$

Any choice of $c_0 > 0$ is equivalent up to rescaling $K_b$ and $\alpha$ via the global constant $\kappa = V/(8c_0)$. In practice we set $c_0 = 1/4$ as a conservative, well–conditioned choice. Thus the change from $1/8$ to $1/4$ in Eq. (11) is not a new assumption, but an instance of this calibration freedom.

**(c) Why division by $r_t$ is required.** Let $M$ denote the sequence length (number of token positions) and let $r_t \in (0, 1]$ be the *coverage ratio* at step $t$, i.e., the fraction of positions that are unmasked and actually subject to classifier guidance. Let $\mathcal{M}_t \subset \{1, \ldots, M\}$ be the edited index set with $|\mathcal{M}_t| = r_t M$. For each token position $i$, we write $s_{t,i} \geq 0$ for its per–token volatility contribution (obtained by aggregating the vocabulary–level gradient at that position).

We first define the mean volatility over *all* positions

$$I_{\text{all},t} := \frac{1}{M} \sum_{i=1}^{M} s_{t,i}, \tag{27}$$

then we can get the mean volatility over the *edited* positions, which is the ideal control signal we want to track.

$$I_{\text{edited},t} := \frac{1}{|\mathcal{M}_t|} \sum_{i \in \mathcal{M}_t} s_{t,i}. \tag{28}$$

Since budget calculation is done after all editing and masked positions do not receive classifier guidance and thus do not contribute to the volatility, we may rewrite

$$I_{\text{all},t} = \frac{1}{M} \sum_{i \in \mathcal{M}_t} s_{t,i} = \frac{|\mathcal{M}_t|}{M} \cdot \frac{1}{|\mathcal{M}_t|} \sum_{i \in \mathcal{M}_t} s_{t,i} = r_t I_{\text{edited},t}. \tag{29}$$

Rearranging equation 29 gives

$$I_{\text{edited},t} = \frac{I_{\text{all},t}}{r_t}. \tag{30}$$

By construction, $I_{\text{all},t}$ is the average volatility over the full sequence and is therefore proportional to our sequence–level volatility $g_{2,t}$ used in the main text; any global proportionality constant can be absorbed into the calibration constant $c_0$. Thus, up to a global scale factor,

$$I_{\text{all},t} \propto g_{2,t} \quad \Rightarrow \quad I_{\text{edited},t} \propto \frac{g_{2,t}}{r_t}. \tag{31}$$

Equation 31 shows that $g_{2,t}$ is a *coverage–diluted* statistic: if we were to use $\lambda_t^2 g_{2,t}$ directly as our control signal, its scale would shrink whenever $r_t$ becomes small, even if the actual edit on the modified tokens is very strong. To obtain a coverage–invariant signal for budget control, we must therefore undo this dilution and work with a quantity proportional to $I_{\text{edited},t} = g_{2,t}/r_t$.

Using the local KL bound in Eq. equation 17, the per–step KL cost is upper–bounded (up to a global constant) by a term proportional to $\lambda_t^2 I_{\text{edited},t}$. Thus any surrogate that is monotone in $\lambda_t^2 I_{\text{edited},t}$ is sufficient for budget control. We choose the simplest calibrated form

$$b_t^{\text{KL}} := c_0 \lambda_t^2 I_{\text{edited},t} \approx \frac{c_0 \lambda_t^2 g_{2,t}}{r_t}, \qquad c_0 = \tfrac{1}{4}, \tag{32}$$

which is exactly Eq. (5) in the main text.

In summary, division by $r_t$ is not an ad–hoc heuristic. It is the formally required normalization that converts the diluted global statistic $I_{\text{all},t} \propto g_{2,t}$ into the true per–edited–token intensity $I_{\text{edited},t}$, so that the accumulated surrogate $\sum_t b_t^{\text{KL}}$ reflects the strength of the applied edits in a coverage–invariant way.

## H   VOLATILITY COMPARISON OF STATIC SCHEDULING AND BALE

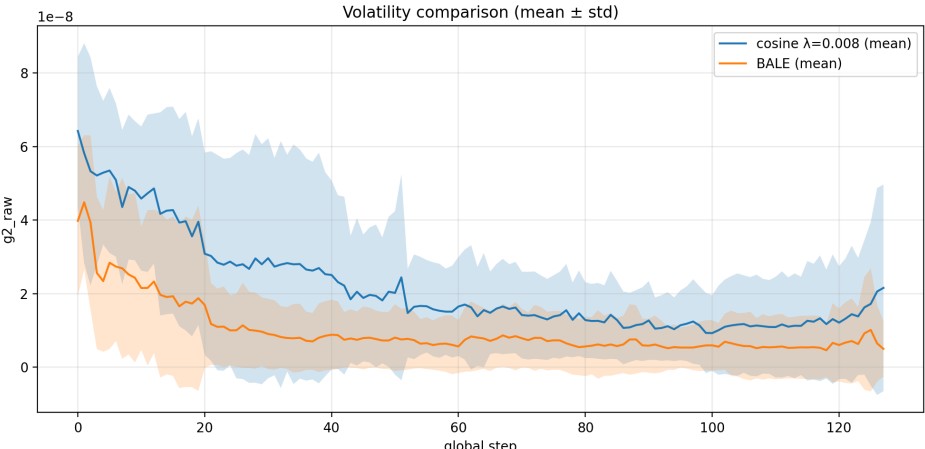

Figure 4: Volatility curve comparision between static scheduling and BALE.

Figure 4 provides a direct comparison of the raw volatility observed during decoding under (i) cosine-$\lambda$ scheduling and (ii) our BALE controller. As shown, the static scheme exhibits fluctuations and several sharp spikes, indicating unstable guidance dynamics. In contrast, BALE rapidly stabilizes the volatility and maintains a consistently lower variance throughout the diffusion process. This confirms that BALE effectively suppresses gradient spikes and delivers smoother, more stable logit updates, supporting our motivation for online control.

## I   ADDITIONAL ABLATION ON DESIGN CHOICE

In this section, we provide additional ablations on $key \oplus neg$ task to clarify which components of the BALE controller are responsible for the performance gains reported in the main paper. We focus on two aspects: (i) the interaction between the residual-based schedule $\lambda_{\text{target}}$ and the gradient rescaling cap $\lambda_{\text{cap}}$, and (ii) the effect of dynamic block-level budget reallocation.

**Schedule vs. gradient rescaling.**   Table 10 compares BALE against variants that only use partial components, as well as static cosine/linear schedules with and without gradient capping($\lambda_{\text{cap}}$). Replacing static schedules with gradient caps does *not* improve cosine or linear guidance—indeed, it often degrades success—whereas combining the cap with our residual-based schedule significantly improves controllability. This supports our claim that the main gains come from the principled, state-aware controller $\lambda_{\text{target}}$, with $\lambda_{\text{cap}}$ acting as a stabilizer rather than a universal trick.

Table 10: Ablation on guidance schedule and gradient rescaling. "Cap only" uses BALE's stability cap $\lambda_{\text{cap}}$ without the KL budget cap, while "w/o cap$_B$, cap" uses only the residual-based schedule $\lambda_{\text{target}}$ without any caps.

| Method | Acc. (Success) | $R_{\text{PPL}}$ |
|---|---|---|
| BALE ($\lambda_{\text{cap}}$ Only) | 0.8600 | 18.787 |
| BALE w/o $\lambda_{\text{capB}}$, $\lambda_{\text{cap}}$ | 0.8450 | 18.953 |
| Cosine + grad cap | 0.6550 | 18.510 |
| Cosine | 0.7350 | 19.010 |
| Linear + grad cap | 0.6775 | 18.240 |
| Linear | 0.6900 | 18.940 |

**Dynamic reallocation across blocks.**   Table 11 isolates the effect of dynamic block-level budget reallocation (Eq. 7). Removing dynamic reallocation causes a large drop in success rate and a substantial degradation in fluency. This confirms that reallocation of remaining budget across blocks is not a cosmetic option but a key part of making the controller stable at block boundaries.

Table 11: Ablation on dynamic block-level budget reallocation (Eq. 7). Results are averaged over 3 seeds (mean $\pm$ std). Dynamic reallocation substantially improves both controllability and fluency.

| Method | Acc. (Success) | $R_{\text{PPL}}$ |
|---|---|---|
| BALE w/ Dynamic Alloc | $0.8750 \pm 0.0153$ | $18.756 \pm 0.293$ |
| BALE w/o Dynamic Alloc | $0.8060 \pm 0.0083$ | $21.195 \pm 0.173$ |

## J    HYPERPARAMETER ROBUSTNESS & SENSITIVITY

We next investigate how sensitive BALE is to its main hyperparameters. Overall, we find that the controller is robust across a wide range of settings: the global KL budget $B$, the blockwise front/back-loading parameter $\rho$, and the stability-related parameters $(\beta, \gamma)$. This supports our claim that BALE does not rely on brittle, narrowly-tuned constants.

**Global KL budget $B$.**    Table 12 sweeps the global budget $B$. When $B$ is extremely small, the controller is overly constrained and success drops. Once $B$ is "loose enough" (roughly $B \geq 5$), performance quickly plateaus and becomes insensitive to the exact value, which is consistent with the interpretation of $B$ as a safety rail rather than a delicate knob.

Table 12: Sensitivity to global KL budget $B$. Performance is robust for $B \geq 5$, forming a plateau in Acc. and $R_{\text{PPL}}$.

| $B$ | 0.1 | 0.5 | 1.0 | 5.0 | 10.0 | 20.0 |
|---|---|---|---|---|---|---|
| Acc. | 0.8425 | 0.8525 | 0.8510 | 0.8725 | 0.8750 | 0.8730 |
| $R_{\text{PPL}}$ | 18.05 | 19.03 | 19.43 | 19.51 | 19.54 | 19.52 |

**Block-level front/back-loading $\rho$.**    Table 13 shows a sweep over $\rho$, which controls how aggressively BALE spends budget early vs. late within a block. Accuracies remain in a narrow band between $0.865$ and $0.880$, indicating that $\rho$ mainly affects the *style* of spending rather than the final outcome.

Table 13: Sensitivity to block-level front/back-loading parameter $\rho$. BALE remains stable across a wide range of $\rho$.

| $\rho$ | 0.25 | 0.50 | 0.75 | 1.00 | 1.25 | 1.50 |
|---|---|---|---|---|---|---|
| Acc. | 0.8750 | 0.8725 | 0.8800 | 0.8750 | 0.8650 | 0.8650 |
| $R_{\text{PPL}}$ | 18.86 | 19.30 | 19.01 | 19.35 | 19.35 | 19.41 |

**EMA decay $\beta$ and cap factor $\gamma$.**    Finally, Table 14 and Table 15 report sensitivity to the EMA decay $\beta$ in the volatility tracker and the scaling factor $\gamma$ used in the stability cap $\lambda_{\text{cap}}$. Both parameters exhibit very flat response curves, confirming that BALE does not depend on precise tuning of EMA dynamics or cap scaling.

Table 14: Sensitivity to EMA decay $\beta$ used in the volatility tracker. Results indicate $\beta$ is not tuning-critical.

| $\beta$ | 0.55 | 0.65 | 0.75 | 0.85 | 0.95 |
|---|---|---|---|---|---|
| Acc. | 0.8730 | 0.8730 | 0.8730 | 0.8760 | 0.8750 |
| $R_{\text{PPL}}$ | 18.71 | 18.71 | 18.74 | 18.77 | 18.86 |

Table 15: Sensitivity to stability cap factor $\gamma$ that scales $\lambda_{\text{cap}}$. BALE is robust over a broad range of $\gamma$.

| $\gamma$ | 1.0 | 1.5 | 2.0 | 2.5 | 3.0 |
|---|---|---|---|---|---|
| Acc. | 0.8760 | 0.8750 | 0.8750 | 0.8760 | 0.8750 |
| $R_{\text{PPL}}$ | 18.75 | 19.02 | 18.86 | 18.82 | 18.99 |

# K  COMPUTATIONAL OVERHEAD

One potential concern is that BALE's more sophisticated controller might introduce prohibitive computational cost compared to static schedules. To quantify this, we measure wall-clock time and peak GPU memory on a single A100 GPU when generating 100 samples with LLaDa-8B.

Table 16 shows that BALE incurs only a small overhead compared to static schedules. The additional latency is roughly $+0.2$ seconds per sample, and the peak VRAM increase is about $+0.27$ GB. In other words, BALE provides substantially better controllability at a modest and practical runtime cost.

Table 16: Wall-clock and memory overhead of BALE compared to static guidance schedules. Measured on a single A100 GPU with 100 generated samples.

| Method | Time / sample (s) | Peak VRAM (GB) |
|---|---|---|
| BALE (Ours) | 7.10 | 19.83 |
| Constant schedule | 6.78 | 19.56 |
| Linear schedule | 6.81 | 19.56 |
| Cosine schedule | 6.90 | 19.56 |
| Prompt-only (no guide) | 4.28 | 16.45 |

# L  EXTENSION TO AUTOREGRESSIVE MODELS

Although BALE is primarily designed for blockwise discrete diffusion models, the controller itself is agnostic to the underlying generative architecture. To illustrate this, we apply BALE to an autoregressive (AR) backbone, Mistral-7B, and compare it against prompt-only baseline and DeAL on the same AR model.

As shown in Table 17, BALE achieves substantially higher constraint satisfaction than DeAL and the prompt-only baseline on the combined constraint setting. However, this comes at the cost of increased $R_{\text{PPL}}$, reflecting that BALE has not yet been optimized for AR decoding. We view this as evidence that the controller is portable across architectures, and as a promising direction for future work in AR-specific adaptation.

Table 17: Applying BALE to an autoregressive backbone (Mistral-7B). BALE achieves substantially higher constraint satisfaction than DeAL and prompt-only guidance, at the cost of higher $R_{\text{PPL}}$.

| Method | Acc. (Success) | $R_{\text{PPL}}$ |
|---|---|---|
| Mistral-7B + PO | 0.6850 | 19.528 |
| Mistral-7B + DeAL | 0.7300 | 20.790 |
| Mistral-7B + BALE | 0.8475 | 30.640 |

