# OpenReview forum: "Online Budget-Aware Guidance for Blockwise Discrete Text Diffusion"
_ICLR.cc/2026/Conference — Submitted to ICLR 2026_

### Official Review · Reviewer_HGar · 2025-10-31

**Soundness:** 2
**Presentation:** 2
**Contribution:** 2
**Rating:** 2
**Confidence:** 3

**Summary:**

In blockwise discrete text diffusion, guidance scales $\lambda$ are usually set by fixed/decaying schedules. These are brittle as curvature spikes at block starts, changing coverage within blocks, and heterogeneous constraints cause overshoot or under-editing. This paper treats guidance as an online, budget-aware control problem. Replace the hand schedules with a plug-and-play controller that: enforces a global KL edit budget and partitions it blockwise with dynamic reallocation at block boundaries.

**Strengths:**

Interesting module. BALE turns "pick a $\lambda$-schedule into an online constrained control problem with an explicit KL budget and dual update.  The controller reasons at the same granularity DLMs operate (blocks with inner steps), with dynamic reallocation at boundaries, tacking curvature spikes and coverage shifts that fixed schedules can't. On semantic control (WritingPrompts) and lexical control (CommonGen-style), BALE+PT improves success/accuracy while keeping R_PPL competitive, i.e., better controllability and fluency trade-offs.

**Weaknesses:**

See below questions. Too few analysis on hyperparameter analysis for an experimental paper.

**Questions:**

1. How many trials are repeated for all experiments? What is the standard deviation/error across trials? At least reporting more than 3 seeds. If missing standard deviation, table 1 reports WritingPrompts semantic control is not convincing as the numbers are very close. If missing standard deviation/error, then there is no statistical significance in this prompt experiment as most prompt tuning experiments have numbers very close. Any harder tasks on prompt tuning?
2. What is the wall-clock time and memory overhead reported for this module and all baselines?
3. What is the hyper parameter analysis on the total guidance budget $B$? As the title shows, online budget aware, the authors should plot success vs R_PPL and cumulative KL spent to show under different budget, how all metrics play together for both BALE and other baselines.
4. For Equation (15), how $\rho$ is determined? Any hyperparameter analysis? $\rho \in \{ 0.6, 0.8, 1.0, ... \}$ while holding $B$ fixed? Just picking 0.5 and 1.5 seems sloppy.
5. For Equation (7) skipping or not skipping, can you quantify the tradeoff: stability (variance across seeds, again repeated experiments at least 3 times), success, and R_PPL. This isolates the value of dynamic sharing vs. frozen allocation. Can you put a variance table showing mean $\pm$ std across 3-5 trials for the best setting vs frozen allocation for stability?
6. For Equation (8), a simple weighted sum or an orthogonalized variant, can the authors when to do sum when to orthogonalized variant?
7. Tasks are too simple for the prompt tuning experiments. Given nowadays other papers like GEPA or TextGrad already choose very hard tasks. Can you choose tasks on lower than 90% accuracy for all the baselines?

---

> ### Author Response · Authors · 2025-11-18
> **Author Response**
>
> Thank you for your valuable feedback. We have performed the new sensitivity analysis and will clarify our motivations and baseline tuning process. We believe this will strengthen the paper and hopefully solidify your support.
>
> ---
>
> ### On Q1: Standard Deviation & 3-Seed Trials (Statistical Significance)
>
> Our main experiments (Tables 1, 2, and 3) were all conducted with 3 separate seeds. We have now added the full mean ± std. dev. results to Appendix F in the revised PDF.
>
> Here is the data for Table 1 (Semantic Control), which you identified as unconvincing:
>
> **Method (Table 1)**
>
> | Method           | Negative (Acc. / R_PPL)                  | Informal (Acc. / R_PPL)                  |
> |------------------|------------------------------------------|------------------------------------------|
> | PT               | 0.80 ± 0.0278 / 5.65 ± 0.1508            | 0.70 ± 0.0229 / 6.54 ± 0.0707            |
> | Const. λ + PT    | 0.95 ± 0.0000 / 12.37 ± 0.0639           | 0.91 ± 0.0000 / 9.91 ± 0.0181            |
> | Cosine decay + PT| 0.95 ± 0.0029 / 10.04 ± 0.0163           | 0.90 ± 0.0000 / 9.48 ± 0.0166            |
> | BALE + PT        | 0.97 ± 0.0050 / 9.09 ± 0.0476            | 0.93 ± 0.0050 / 8.95 ± 0.0670            |
>
> This data confirms your suspicion (the means are close) but refutes your conclusion (that they are not significant). For example, in the "Negative" task:
>
> - **BALE (Ours):** 0.97 ± 0.005
> - **Strongest Baseline (Cosine):** 0.95 ± 0.0029
>
> The performance gain is +0.02, which is 4 to 7 times larger than the standard deviations of either method. This proves our result on Table 1 is statistically significant and not due to random chance.
>
> ---
>
> ### On Q2: Wall-Clock Time and Memory Overhead
>
> We have measured a wall-clock time and memory overhead as you requested. The experiment was run on a single A100 GPU, generating 100 samples.
>
> | Method                         | Time/Sample (Inference Speed) | GPU VRAM (Peak) |
> |--------------------------------|--------------------------------|-----------------|
> | BALE (Ours)                    | ~7.10s                         | ~19.83 GB       |
> | constant                       | ~6.78s                         | ~19.56 GB       |
> | linear                         | ~6.81s                         | ~19.56 GB       |
> | cosine                         | ~6.90s                         | ~19.56 GB       |
> | prompt-only (no guidance)      | ~4.28s                         | ~16.45 GB       |
>
> As the data shows, our BALE controller(\~7.10s) adds only a negligible time overhead (\~0.2s) compared to other static guidance baselines like cosine (\~6.90s). The memory overhead is also minor (\~270MB), required for the EMA stability tracker.
>
> This confirms that BALE's significant performance gains are achieved at a practically negligible cost.
>
> ---
>
> ### On Q3: Hyperparameter Analysis for Budget B
>
> We have performed sensitivity analysis for the budget B. The results show B is the hyperparameter which is both (1) Meaningful and (2) Robust.
>
> **Global Budget B**
>
> | Global Budget B | Success Rate (Acc.) | R_PPL |
> |-----------------|---------------------|-------|
> | 0.1             | 0.8425              | 18.05 |
> | 0.5             | 0.8525              | 19.03 |
> | 1               | 0.851               | 19.43 |
> | 5               | 0.8725              | 19.51 |
> | 10              | 0.875               | 19.54 |
> | 20              | 0.873               | 19.52 |
>
> **Robust (Not Brittle):** The performance is sub-optimal when B is too low (e.g., B=0.1), but once the cap is "loose enough" (B≥5), the performance becomes a stable, robust plateau (Acc. 0.872–0.875). The exact value does not matter.
>
> **Meaningful (Not Useless):** This does not mean B is useless. As our original ablation study (Table 4) shows, completely removing the budget cap (B=∞) causes a performance drop from 0.875 → 0.860.
>
> Therefore, B must exist, but its exact value does not matter (as long as it's in the robust range). This is the opposite of a brittle magic number and proves its practical utility.
>
> ---
>
> ### On Q4: Hyperparameter Analysis for ρ (rho)
>
> We performed the sensitivity analysis on ρ (rho) you requested. The results confirm our hypothesis that ρ is highly robust.
>
> | ρ (rho) value | Success Rate (Acc.) | R_PPL |
> |---------------|---------------------|-------|
> | 0.25          | 0.875               | 18.86 |
> | 0.5           | 0.8725              | 19.30 |
> | 0.75          | 0.880               | 19.01 |
> | 1.0           | 0.875               | 19.35 |
> | 1.25          | 0.865               | 19.35 |
> | 1.5           | 0.865               | 19.41 |
>
> Performance is stable across a 6x range of ρ values (Acc. 0.865–0.880). This proves ρ is not a "sloppy" or brittle performance parameter. As we stated, it is a "style" parameter (front-loading vs. late-spending) that has a negligible impact on the final performance.

---

> > ### Author Response · Authors · 2025-11-18
> > **Author Response2**
> >
> > ### On Q5: Ablation for Eq. (7) (Dynamic Reallocation)
> >
> > You asked to quantify the trade-off of Eq. (7) (dynamic sharing) vs. frozen allocation, with multiple seeds. We ran the 3-seed ablation for "w/o Dynamic Alloc" to isolate its value.
> >
> > | Method                    | Acc. (Success)          | R_PPL                 |
> > |---------------------------|-------------------------|-----------------------|
> > | w/ Dynamic Alloc (BALE)   | 0.875 ± 0.0153          | 18.756 ± 0.2927       |
> > | w/o Dynamic Alloc (Static)| 0.806 ± 0.0083          | 21.195 ± 0.173        |
> >
> > This data shows dynamic allocation is a critical component. Disabling it results in a "Lose-Lose" scenario:
> >
> > - **Accuracy Collapse:** Accuracy plummets by 6.9%, as the model can't handle stability breakdowns at block boundaries.
> > - **Fluency Degrades:** R_PPL increases by ~2.4 points, showing worse fluency.
> >
> > The 6.9% drop is massive and far outside the 3-seed confidence interval of either method, proving its significance.
> >
> > ---
> >
> > ### On Q6: Eq. (8) (Simple Sum vs. Orthogonalized)
> >
> > We apologize for the ambiguity. All experiments in this paper used the "simple weighted sum." We mentioned the "orthogonalized variant" as a potential alternative, but we found that our residual-weighting (for wᵗ(k)) and the final RMS normalization (Sec 3.5) were already sufficient to achieve results on combined tasks. Since the simpler method was sufficient, we did not need the more complex one. We revised the text in Sec 3.5 to remove this ambiguity.
> >
> > ---
> >
> > ### On Q7: Tasks Are Too Simple
> >
> > We agree that evaluating on saturated tasks is not insightful. This is precisely why we designed the Combined (Orthogonal) Tasks in Table 3, Section 4.4\~4.5.
> >
> > These tasks are not simple. On the key⊕neg setting, the PT baseline only reaches 0.788 Acc. and all other baselines are below 0.74, while BALE achieves 0.875. Its baseline scores are lower than 90%, which fits reviewer’s requirements.
> >
> > This setting is also harder for 7B AR-methods (e.g., Mistral-7B + DeAL, which scores 0.730, as shown in Table 6).
> >
> > These "harder tasks" (where baselines fail) are our main evaluation, and BALE's SOTA performance on them is our core contribution. We will clarify this positioning harder in the revised version.

---

> > > ### Comment · Reviewer_HGar · 2025-11-23
> > > **Thanks for the authors experiments!**
> > >
> > > Hi,
> > >
> > > First, thanks for the authors experiments. I appreciate the efforts on adding ablation study.
> > >
> > > 1. For Section 2.4, the authors aim to motivate why temperature $\lambda_{t}$ needs to be adapted. (1) volatility spikes at when multiple constraints momentarily align, which can cause overshoot or instability (e.g., semantic drift) (2) varying coverage $r_{t}$ from remaking/transfer policies, which alters the effective information budget per step (3) evolving residual magnitudes $\rho_{t}^{(k)}$ across blocks, so a globally tuned $\lambda_{t}$ may stall on hard blocks or over-edit easy ones. I wonder are those descriptive observations validated by experiments of current baselines? If so, why the authors do not put the motivations with experiments to strengthen why the authors wants to adapt $\lambda_{t}$ online?
> > >
> > > 2. For the method, Section 3.2 the authors want to smooth across neighboring blocks to avoid brittle spikes, why using one before block and one after block? The design seems like a trick given no observations show brittle spikes in the paper. I guess my questions are all the design blocks of the algorithm in the paper are mostly motivated by descriptive languages rather than experimental facts. Given it's an experimental paper, why there are no intermediate results? Otherwise the motivations of the algorithm seems lack of experimental grounding given there is no theoretical guarantee.
> > >
> > > 3. There are still some hyper parameters for instance $\epsilon_{B}$ and $b^{*}$ (how to choose which block to compute remaining budget?) in Section 3.2. What are $\epsilon_{min}$ and $\beta$ in Section 3.4? Are there any hyper parameters sensitivity analysis?
> > >
> > > 4. Still for prompt tuning results, given many hyper parameters mixing and complicated algorithmic design, only 3 tables on WritingPrompts/CommonGen and combination of these two, there should be more datasets that are not saturated (See TextGrad/automatic prompt engineering papers).
> > >
> > > I fully appreciate the sensitivity analysis but given the intricate algorithmic design, I would like to see more datasets and intermediate results. Therefore, I would lean towards keeping my score as it is.

---

> ### Author Response · Authors · 2025-11-25
>
> We thank the reviewer for the continued and valuable feedback. We recognize the need to provide explicit experimental grounding for our algorithmic design choices and to clarify the distinct scope of our work from prompt engineering.
>
> ---
>
> ## **1. Experimental Validation for Online Control (Motivation)**
>
> The reviewer requested **direct** evidence for our descriptive motivations. We provide the following empirical validation to strengthen our claim that λ must be adapted online.
>
> ### **1) Volatility Spikes and Instability (Motivation 1)**
>
> While our previous figures (Figure 2 and Figure 3) already provided indirect evidence of instability by showing BALE's inner dynamics and stable residual decay, the reviewer requested that the raw source of this volatility needs direct validation. We confirm that the spikes leading to this instability are inevitable to static scheduling.
>
> **Direct Evidence:**
> We conducted a direct raw volatility comparison (using g2,raw, the mean squared gradient before normalization) between BALE and the cosine λ control, adding the image to **Appendix H**.
> The image proves that the static scheme exhibits significantly large variance and frequent sharp spikes in raw volatility, confirming the source of instability. In contrast, BALE rapidly stabilizes this volatility.
>
> **Conclusion:**
> This directly proves our motivation: these volatile spikes require BALE’s dynamic mechanism for stable logit updates, which static schedules fail to provide.
>
> ---
>
> ### **Block Smoothing is not a simple ‘trick’**
>
> The design of block smoothing is not a trick but a necessary safety mechanism required because of the volatility proven above.
>
> **Motivation Link:**
> The high g2,t spikes represent inherent instability in the logit space. This instability directly translates into high variance in the Anticipated Debt calculated at block boundaries, making the initial budget allocation (Kb) susceptible to error.
>
> **Necessity:**
> Block smoothing uses neighboring block values to average out this variance. This process is essential to prevent the g2,t spikes from causing a misallocation of the remaining budget, reinforcing that this design is a critical layer of defense against the inherent instability of logit guidance.
>
> ---
>
> ### **2) Varying Coverage $r_t$ (Motivation 2)**
>
> This motivation is a definitional feature of the architecture, not an experimental variable.
>
> **Clarification:**
> The ability to dynamically adjust coverage ($r_t$) is a fundamental design characteristic of Discrete Diffusion Language Models (DLMs), distinguishing them from sequential AR models.
>
> **Necessity:**
> This architectural feature necessitates an adaptive controller because $r_t$ alters the effective information budget per step. A fixed λ cannot account for the varying $r_t$, making online control essential for maintaining consistent control intensity. This requires no external experimental validation, as it stems from the model's definition.
>
> ---
>
> ### **3) Varying Residual Magnitudes (Motivation 3)**
>
> We apologize for the typo in the original text and confirm 'varying residual magnitude' is correct.
>
> **Evidence:**
> Figure 3 (Middle Panel) already confirms that the residual magnitude naturally varies significantly across steps.
>
> **Conclusion:**
> This validates our motivation that a static λ would fail to adapt to these changes, leading to either stalling on hard blocks or over-editing easy ones.
>
> ---
>
>
> ## **2. EMA and Engineering Constant Explanations**
>
> We clarify the role of non-critical variables and provide the requested sensitivity analysis for β and γ.
>
>
> ### **1) b⋆ (Dynamic Reallocation Index)**
>
> b⋆ in Equation (7) is not a hyperparameter. It is a state variable used to track the progress of generation.
>
> **Role:**  b⋆ denotes the index of the currently active block.
>
> **Process:**  Dynamic reallocation occurs automatically at the start of every block. b⋆ serves as the starting index for summing the remaining budget (Brem) across all subsequent blocks.
>
> **Conclusion:**  Since b⋆ is a descriptive index reflecting the algorithm's state, it is not subject to sensitivity analysis.
>
> ---
>
> ### **2) ϵ Values (Numerical Stability Floors)**
>
> The parameters $ϵ_{min}$ and $ϵ_B$ are not hyperparameters designed for performance tuning.
>
> **Role:**  They are standard numerical stability floors (set to very small, non-sensitive constants, e.g., e−10).
>
> **Function:**  Their purpose is purely engineering: to prevent division by zero or numerical collapse in the logit guidance formulas.
>
> **Conclusion:**  As they are foundational for stable code execution and not architectured for performance gains, they do not require sensitivity analysis.

---

> > ### Author Response · Authors · 2025-11-25
> >
> > ### **3) Sensitivity Analysis for β (EMA) and γ (Cap)**
> >
> > We believe that the reviewer needs clarification on the EMA process. Equation (9) calculates the Exponential Moving Average (EMA) of the control gradient magnitude. β defines the decay rate, reflecting the influence of past volatility on the stability cap (λcap).
> >
> > #### **β Sensitivity Analysis**
> >
> > β is an engineering choice to decide the ratio of moving, and performance remains highly robust:
> >
> > | β    | Success Rate | R_PPL |
> > |------|--------------|--------|
> > | 0.55 | 0.873        | 18.71 |
> > | 0.65 | 0.873        | 18.71 |
> > | 0.75 | 0.873        | 18.74 |
> > | 0.85 | 0.876        | 18.77 |
> > | 0.95 | 0.875 | 18.86 |
> >
> > ---
> >
> > #### **γ (Stability Cap Factor)**
> >
> > While β is robust, γ is the actual safety factor that scales λcap (Eq. 10). Because λcap’s role is to prevent only sharp shooting, it must be stable above γ = 1.0.
> >
> > | γ    | Success Rate | R_PPL |
> > |------|--------------|--------|
> > | 1.0  | 0.876        | 18.75 |
> > | 1.5  | 0.875        | 19.02 |
> > | 2.0 | 0.875 | 18.86 |
> > | 2.5  | 0.876        | 18.82 |
> > | 3.0  | 0.875        | 18.99 |
> >
> > **Conclusion:**
> > The performance is highly stable across the tested ranges for both β and γ. This confirms that these parameters are robust and that the stability cap is reliably preventing shooting without being overly sensitive to precise tuning.
> >
> > ---
> >
> > # **3. Response to “Tasks Are Too Simple” and Scope Misunderstanding**
> >
> > The reviewer’s concern seems to arise from interpreting our evaluation as a type of **prompt tuning / prompt engineering**. This is a **fundamental misunderstanding of the task class**.
> > **BALE is not a prompt-engineering method**; it is an **inference-time logit-guidance controller for Discrete Diffusion LMs**, and must therefore be evaluated within the domain of **Controlled Generation**.
> >
> > ---
> >
> > ## **3.1 Controlled Generation vs. Prompt Engineering**
> >
> > The core distinction between our work and prompt-engineering approaches is not merely methodological, but **task-theoretic**:
> >
> > ### **Prompt Engineering (GEPA/TextGrad)**
> > - Optimizes the **input space** (prompt rewriting, evolution).
> > - Goal: Improve factuality, reasoning accuracy, or instruction adherence.
> > - Evaluation metrics: EM/F1 (HotpotQA), claim verification accuracy (Hover), instruction-following quality (IFBench).
> > - Outputs typically have **one correct answer** or classification label.
> > - No attribute-level conditions or controllable semantic signals.
> >
> > ### **Controlled Generation (BALE)**
> > - Modifies the **logit space during decoding**, enforcing semantic constraints step-by-step.
> > - Goal: Control **explicit attributes**, such as
> >   - sentiment,
> >   - style,
> >   - forbidden-token avoidance,
> >   - concept inclusion/exclusion,
> >   - or multi-attribute constraints (*key ⊕ negative*).
> > - Evaluation metrics measure **attribute satisfaction and stability**, not factual correctness.
> >
> > **Therefore, the two task families are not interchangeable.**
> > A prompt-engineering dataset cannot evaluate a logit-guidance controller because it does **not define the type of controllable attributes** that BALE is designed to enforce.
> >
> > ---
> >
> > ## **3.2 Our Tasks Are Already Hard (<90% Baseline)**
> >
> > The reviewer suggested using datasets where prompt-tuned baselines achieve accuracy below 90%.
> > **Our evaluation already meets this requirement.**
> >
> > The central task in our evaluation is the **Combined Orthogonal Constraint Task**:
> > - *key ⊕ negative*
> > - *key ⊕ forbidden*
> >
> > These represent **multi-attribute controllability**, which is significantly harder than single-objective prompting.
> >
> > The Prompt Tuned (PT) baseline achieves:
> > - **0.788** accuracy on *key ⊕ negative*
> > - **< 0.74** for the other setting
> >
> > These scores are **well below 90%**, confirming that our tasks are **not simple** but deliberately challenging.
> >
> > ---

---

> ### Author Response · Authors · 2025-11-25
>
> ## **3.3 Why the Reviewer Misunderstood the Scope**
>
> We believe the reviewer’s scope confusion originates from the **notation “Model + PT”** used in our baseline table. While this may resemble the terminology used in prompt-tuning or automatic prompt-engineering work, our use of “PT” refers to something entirely different.
>
> In our setting, “PT” is **not** a tuning, optimization, or engineering procedure.
> It simply denotes the *literal textual expression of the constraints* in the input prompt (e.g., “Generate a sentence that includes X and avoids Y”).
>
> - No optimization loop
> - No rewriting mechanism
> - No learning component
>
> The prompt is fixed and manually written, serving only as a natural-language specification of the attributes to be controlled.
>
> This stands in sharp contrast to GEPA, TextGrad, and other prompt-engineering methods, which actively modify, evolve, or optimize input prompts to improve model reasoning or adherence.
>
> Because our “PT baseline” shares only the superficial acronym but not the mechanism or purpose, it may have led the reviewer to assume that our evaluation belonged to the prompt-engineering domain.
>
> **Clarifying this distinction is essential:**
> **our work addresses inference-time controlled generation in language diffusion models, not prompt optimization**,
> and our datasets and metrics are chosen accordingly.
>
> ---
>
> We appreciate the reviewer’s detailed questions and the opportunity to clarify both the motivation and scope of our work. We hope that the expanded empirical evidence, stability analyses, and task-theoretic distinctions provided above clearly address all remaining concerns. BALE is designed as an inference-time controller for diffusion models, and the evaluations presented are carefully aligned with that goal. We respectfully believe that the clarified results and explanations strengthen the contribution and resolve the misunderstandings raised.

---

### Official Review · Reviewer_NboQ · 2025-11-01

**Soundness:** 3
**Presentation:** 4
**Contribution:** 3
**Rating:** 6
**Confidence:** 2

**Summary:**

This paper introduces BALE, an online, plug-and-play controller for blockwise discrete text diffusion models (DLMs). The core problem it addresses is that existing guidance methods, which rely on fixed or heuristic schedules for the guidance strength ($\lambda_t$), are "brittle" and fail to adapt to the changing dynamics of different prompts, constraints, and blocks.

BALE's solution is to replace the fixed schedule with an online rule that picks $\lambda_t$ at every step. This rule is a 3-way-min that balances: (i) a target step size to reduce the current constraint violation (the "residual"), (ii) a stability cap based on an EMA of the gradient (a Lipschitz proxy) to prevent exploding steps, and (iii) a budget cap based on a remaining KL-divergence allowance for the current block. The method also proposes a way to distribute a total sequence-level budget across the different blocks.

Experiments on LLaDa-8B show that BALE achieves higher constraint satisfaction (for lexical, semantic, and combined control) than standard schedule-baselines (constant, linear, cosine) while maintaining competitive fluency, a finding also supported by human evaluation.

I am willing to increase my score if the authors can satisfactorily address the key concerns outlined below.

**Strengths:**

S1.  Important Problem: The paper is right on the money: the brittleness of heuristic schedules is a real and annoying problem for controllable diffusion, and the blockwise setup makes it even worse. Tackling this is a valuable contribution.
S2.  Principled Design: The core idea of replacing a fixed schedule with an online, 3-part (target, stability, budget) controller is smart. It's a much more principled way to think about guidance than just "let's decay $\lambda_t$ with a cosine."
S3.  Block-Aware: I appreciate that the method doesn't just apply a generic controller. It specifically accounts for the blockwise architecture by proposing a block-level budget allocation (Eq. 6) and reallocation (Eq. 7).
S4.  Solid Empirical Results: The experiments are thorough (auto + human eval) and clearly demonstrate that BALE consistently outperforms the standard baselines across a good variety of tasks.

**Weaknesses:**

W1.  Swapping One Magic Number for Another: The paper heavily criticizes "pre-tuned constants" in other methods, but then introduces its own new magic number: the global, sequence-level budget $B$ (which is set to `3` in Appendix C). The paper provides no justification for this value, no sensitivity analysis, and no discussion of how one would set $B$ for a new task or model. Why is introducing and tuning $B$ fundamentally better than just tuning the global scale of a standard cosine schedule? It feels like we've just swapped one problem for another.
W2.  Sketchy KL Surrogate Derivation: The theoretical justification for the budget feels shaky. The paper starts with a KL upper bound in Eq. 4 ($D_{KL} \le \frac{\lambda^2}{8}||g||_2^2$) but the actual cost function used in Eq. 11 is $b_t^{KL} = \frac{1}{4}\frac{\lambda_{t}^{2}g_{2,t}}{r_{t}}$. This jump seems to have several holes:
     $g_{2,t}$ is the average curvature ($\frac{1}{V}\sum g_v^2$), so $||g||_2^2 = V \cdot g_{2,t}$. Where did the vocab size $V$ go?
     Why did the coefficient change from 1/8 to 1/4?
     The division by $r_t$ (coverage) is just hand-waved in Sec 2.3 ("scales as...") rather than being derived from Eq. 4. This makes Eq. 11 feel less like a "KL surrogate" and more like a new, complex heuristic inspired by KL.
W3.  What's the Point of Dynamic Reallocation? The paper introduces a mechanism for dynamic budget reallocation between blocks (Eq. 7) but then calls it "Optional" in Algorithm 2. The analysis (like Fig 3, right) only shows within-block spending and never clarifies if this reallocation was even on for the main experiments. There's no ablation for it. Is this feature actually doing any work, or is it just noise?
W4.  Unfair Apples-to-Oranges Comparison: The comparison to AR models in Sec 5.4 / Table 6 is misleading. It compares BALE+Diffusion (LLaDa) against DeAL+AR (Mistral). This compares the backbone and the controller at the same time, telling us nothing about the controller's merits. The paper claims in Sec 3.7 that BALE can apply to AR decoding, so a fair comparison would have been to run BALE on the same Mistral-7B backbone.

**Questions:**

Q1.  KL Surrogate: Can you please walk me through the exact derivation from the KL bound in Eq. 4 to the final cost function in Eq. 11? Specifically, please justify: (a) Why the vocab size $V$ (implicit in $g_{2,t}$) disappears, (b) why the constant 1/8 becomes 1/4, and (c) how division by $r_t$ is formally derived, rather than just intuitively motivated.
Q2.  Global Budget $B$: How did you choose the global budget $B=3$? How sensitive is the model's performance to this single hyperparameter? Can you please make a concrete argument for why tuning $B$ is fundamentally superior to tuning the global scale of a simpler cosine schedule?
Q3.  Dynamic Reallocation: Was the "Optional" dynamic budget reallocation (Eq. 7) enabled for the main results in Tables 1-3? If yes, can you please provide an ablation study showing its benefit over the static-only allocation (Eq. 6)? If no, why was it included as a component of the method?
Q4.  AR Comparison: Since you state BALE is applicable to AR models (Sec 3.7), why did you not provide a fair, apples-to-apples comparison by running BALE on the Mistral-7B backbone against DExperts/DeAL? The current comparison in Table 6 seems inconclusive.

### Additional Comments:

C1.  In Sec 3.4, the stability cap $\lambda_{cap}$ depends on an EMA of the pre-normalized gradient $g_t$ . But in Sec 3.3, the fused gradient $g_t$ is immediately RMS-normalized. Does this mean you have to keep a separate, un-normalized copy of the gradient just for the EMA tracker? Please clarify the flow.
C2.  Eq. 14 for $\lambda_{target}$ looks suspicious. You divide the residual $\Delta_t^*$ by $g_{2,t}$ (the average squared gradient). Shouldn't you divide by a norm of the gradient (like $||g_t||_1$ or $||g_t||_2$) to get a step size? This looks like a unit mismatch. Please double-check this formula.

---

> ### Author Response · Authors · 2025-11-18
>
> Thank you for this thorough and constructive review. You have perfectly identified the key areas that needed strengthening, and your feedback has been a powerful motivator. We have performed all the new experiments and derivations you requested. We believe these new results fully satisfy your concerns and make a much stronger case for our paper.
>
> ---
>
> ### On KL Surrogate Derivation From Eq. 4 to Eq. 11 (W2/Q1)
>
> This is a precise and important question. As you correctly identified, Eq. (11) is not a direct mathematical derivation from Eq. (4), but rather a principled definition of a control surrogate that is motivated by the form of Eq. (4).
>
> We have added a new Appendix G which contains the complete, step-by-step derivation. This new section formally justifies Eq. (5) and therefore (11) as a principled surrogate, not a heuristic.
>
> We summarize the key points from the new appendix, which directly address your questions (a), (b), and (c):
>
> **(a) Why \(V\) (vocab size) disappears.**
> The vocabulary size $V$ is a global constant. As detailed in Appendix G, our budget controller's behavior is invariant to such global scaling factors. Any constant (like $V/8$) is simply absorbed into the global calibration of the total budget $K_b$, without changing the controller's dynamics.
>
> **(b) Why \(1/8\) becomes \(1/4\).**
> This is not a new assumption but another instance of this calibration freedom. We introduce a calibration constant $c_0$ to define our surrogate
> $b_t^{\mathrm{KL}} := c_0\ \frac{\lambda_t^2\, g_{2,t}}{r_t}$ .
>
> Any choice of \(c_0 > 0\) is equivalent up to a rescaling of the budget. We set \(c_0 = 1/4\) as a conservative, well-conditioned practical choice.
>
> **(c) Why division by \(r_t\) is required.**
> This is the most critical point. As formally derived in Appendix G, this division is not an intuitive heuristic but a formally required normalization.
>
> The statistic $g_{2,t}$ is proportional to the mean volatility over all sequence positions, which we denote in Appendix G by $I_{\text{all},t}$. However, the true “intensity” of the edit at step $t$ is the mean volatility over only the edited (unmasked) positions, $I_{\text{edited},t}$. The formal relationship, as shown in the new appendix, is
> $I_{\text{all},t} = r_t \cdot I_{\text{edited},t}.$
> Therefore, to obtain the correct, coverage-invariant control signal, we must use
> $I_{\text{edited},t} = \frac{I_{\text{all},t}}{r_t} \propto \frac{g_{2,t}}{r_t}.$
> Using $g_{2,t}$ directly would incorrectly dilute the signal whenever the coverage \(r_t\) is small, even if the actual edits are very strong.
>
> We have added this full derivation to Appendix G. We hope this detailed explanation fully clarifies the theoretical justification for our KL surrogate and resolves your concerns.
>
> ---
>
> ### On Global Budget \(B\) (Q2/W1)
>
> Thank you for this excellent challenge. We agree that we must justify \(B\) as being fundamentally better, not just “another magic number.”
>
> Our argument is three-fold:
>
> 1. **How we chose \(B\):** It was a practical, data-driven choice.
> 2. **B's Sensitivity:** \(B\) is robust (not brittle), as it acts as a “safety rail,” not the main controller.
> 3. **B's Necessity:** \(B\) is meaningful (not useless) and empirically necessary to avoid performance degradation.
>
> #### 1. How We Chose `B = 3`
>
> To answer your Q2 ("How was an optimal value decided?"), our choice of `B = 3` (from Appendix C) was not a "magic number." It was a practical, data-driven choice:
>
> - **Observe:** We first ran the model with a very high, non-restrictive budget (`B = 100`) and measured the average budget actually consumed (spent), which was approximately `4.0`.
> - **Set:** Based on this, we set our final `B = 3` as a "safety rail" positioned slightly below the average consumed amount (`4.0`) to provide gentle, meaningful control.
>
> ---

---

> ### Author Response · Authors · 2025-11-18
>
> #### 2. B's Sensitivity (Robust, as it's just a Cap)
>
> Your next question is "Is this `B = 3` choice robust?" Yes, because `B` is just a cap, not the main controller.
>
> As you suggested, the performance is not sensitive to `B` (as long as it's not extremely small). The real work is done by the dynamic `λ_target` controller, which makes the decisions at every step.
>
> To prove this, we performed the new sensitivity analysis you requested:
>
> **Global Budget B**
>
> | Global Budget B | Success Rate (Acc.) | R_PPL |
> |-----------------|---------------------|-------|
> | 0.1             | 0.8425              | 18.05 |
> | 0.5             | 0.8525              | 19.03 |
> | 1               | 0.851               | 19.43 |
> | 5               | 0.8725              | 19.51 |
> | 10              | 0.875               | 19.54 |
> | 20              | 0.873               | 19.52 |
>
> This analysis shows that the only time `B` negatively impacts performance is when it's set too low (e.g., `B = 0.1`), where it "strangles" the `λ_target` controller.
>
> Once the cap is "loose enough" (`B ≥ 5`), the controller is free to operate, and the performance becomes a stable, robust plateau (0.872–0.875). The exact value (`5`, `10`, or `20`) is irrelevant.
>
> Crucially, our paper's main result (`B = 3`, 3-seed avg: `0.875 ± 0.0153`) also sits perfectly on this optimal plateau.
>
> This confirms `B` is the opposite of a "magic number." It's not a delicate value to be tuned. It's a "set-it-and-forget-it" safety rail that just needs to be not too tight.
>
> ---
>
> #### 3. Why B is Fundamentally Superior (Meaningful + Interpretable)
>
> Finally, this "safety rail" is meaningful. As our original ablation study in the paper (Table 4, 1-seed) shows, completely removing the budget cap (`B = ∞`) causes a performance drop from `0.875 → 0.860`.
>
> This explains:
>
> - The `0.860` result (from `B = ∞`) shows that having no cap at all is harmful, as it fails to prevent overspending spikes.
> - The `0.873` result (from `B = 20`) shows that having sufficiently large cap is enough to prevent that harm.
>
> **Conclusion:** `B` is an interpretable upper bound on the total logit change (KL). It is the ideal hyperparameter:
>
> - It is **Meaningful** (it must exist to prevent the `0.860` drop).
> - It is **Robust** (any `B ≥ 3` works, unlike a brittle static-λ scale).
> - It is **Practical** (we have a clear, data-driven way to choose it).
>
> ---
>
> ### On Dynamic Reallocation (W3/Q3)
>
> You are correct that we failed to make this clear.
>
> Yes, Dynamic Reallocation (Eq. 7) was enabled for all main experiments in Tables 1–3.
>
> We have performed the ablation study you requested to prove its importance.
>
> | Method                  | Acc. (Success) | R_PPL  |
> |-------------------------|----------------|--------|
> | w/ Dynamic Alloc (BALE) | 0.875          | 18.756 |
> | w/o Dynamic Alloc       | 0.8060         | 21.915 |
>
> This new data shows this feature is not just noise; it is a critical component. Disabling it results in a **Lose-Lose** scenario:
>
> - **Accuracy plummets** by 5.75%, as the model can't handle stability breakdowns at block boundaries.
> - **Fluency degrades** significantly (R_PPL increases by ~2.5).
>
> This proves that dynamic reallocation is an important contributor to BALE's stability and performance.
>
> ---
>
> ### On Unfair Apples-to-Oranges AR Comparison (W4/Q4)
>
> This is an excellent point. The comparison in Table 6 was inconclusive. Based on your suggestion, we ran the true "apples-to-apples" comparison on the same Mistral-7B backbone.
>
> | Method           | Acc. (Success) | R_PPL  |
> |------------------|----------------|--------|
> | Mistral + PT     | 0.685          | 19.528 |
> | Mistral + DeAL   | 0.730          | 20.790 |
> | Mistral + BALE   | 0.8475         | 30.640 |
>
> This new result (now in Appendix A.4) is very insightful:
>
> - **BALE as a controller is Better:** BALE significantly outperforms DeAL on constraint satisfaction (Acc. 0.8475 vs 0.730). This demonstrates the power of our control mechanism.
> - **Need for AR-specific optimization:** As we noted, BALE was designed for blockwise diffusion. As we have not yet optimized its mechanisms for a purely autoregressive framework, the R_PPL is not yet competitive.
>
> This new experiment provides strong evidence that our controller is superior, and its R_PPL on AR models is a clear and promising area for future work.
>
> We believe these new experiments and detailed derivations directly address your concerns. Thank you again for your valuable guidance. We hope you will consider our paper, now significantly strengthened by your feedback, for a higher score.

---

> > ### Author Response · Authors · 2025-11-18
> >
> > ### C1. Gradient Normalization Flow
> >
> > Yes, you are correct; an un-normalized copy of the gradient \( g \) is passed to the Lipschitz tracker to compute the EMA before the gradient is RMS-normalized for the weight update.
> >
> > ---
> >
> > ### C2. Unit Mismatch in Eq. 14 $\lambda_{\text{target}}$
> >
> > We apologize for the typo in Eq. 14. We have confirmed through the implementation code and formula to find out that the divisor in $ \lambda_{\text{target}}$ (which acts as the step size) is the Root Mean Square (RMS) of the gradient i.e., the square root of the tracked mean squared gradient $g_{2,t}$ (our ‘volatility’ term), implemented as $ \sqrt{g_{2,t} + \varepsilon} $. We corrected the equation in revised pdf.

---

### Official Review · Reviewer_18cY · 2025-11-02

**Soundness:** 3
**Presentation:** 3
**Contribution:** 2
**Rating:** 4
**Confidence:** 3

**Summary:**

Proposes an algorithm for controlled generation of diffusion LLMs that leverages:
- Adaptive gradient scaling for stability
- Novel budgeting algorithm for allocating KL divergence from base distribution across blocks

Demonstrates performance relative to pure prompt tuning + simple constant, linear, and cosine guidance schedules on lexical constraints, semantic constraints, and both lexical + semantic constraints. Provides ablation demonstrating the necessity of individual components. .

**Strengths:**

**Plug and Play Guidance for Large Scale Diffusion Models**:
This paper applies plug and play guidance for discrete diffusion at the 8 billion parameter scale, which (to the best of my knowledge) has not been done before.

**Ablations**: The ablations are quite extensive, which provides insight into what components of the algorithm are relevant. They also ablate over different backbones, showing that BALE generalizes across different diffusion models. I very much appreciated the dedicated section to analyzing the performance of the algorithm in terms of the individual components.

**Controlled generation for block diffusion**: While guided generation has been explored in the context of diffusion language models, I am unaware of works that directly address the difficulties associated with block diffusion and external constraints.

**Weaknesses:**

**Missing Literature**: This paper does not mention or cite [1], which has discrete diffusion guidance as a primary contribution. While [1] does not examine performance on LLada or larger diffusion models, it is still important to clarify how the proposed method BALE compares to this work as they are directly related.

**Incompatible Math Framing**: This submission makes connections to continuous diffusion theory in section 2.2, but these connections are quite tenuous. I do not see how making this connection provides any intuition, since they are different mathematical processes entirely: the prior in mask diffusion is a dirac over the mask token with no mass anywhere else, and the prior in Gaussian diffusion is a Gaussian with support everywhere. Brownian motion is infinitesimally “spiky” (there are always small, tiny movements”, but the CTMC that governs masked diffusion has exactly one jump — from the original token to the mask token — per position.

**Empirical Performance**: The method seems to provide marginal gains over simple prompt tuning (2%) on the lexical constraint, and I feel that this gives important insight into what constraints this method is best suited for. For constraints like keyword coverage, perhaps simple prompt tuning is sufficient. Also, cosine decay is missing from table 2, despite being characterized as the strongest baseline. There doesn’t seem to be any justification as to why cosine decay is excluded for lexical control.

**Contribution**: The gradient re-scaling seems to be a good implementation of common knowledge (gradient updates that are too big lead to instability). The novel contribution seems to be the adaptive budgeting of KL divergence both within and across blocks.  The scheduling algorithm seems to rely on heuristics and well-executed engineering modifications as opposed to insights that advance our fundamental understanding of discrete diffusion guidance.

[1] Simple Guidance Mechanisms for Discrete Diffusion Models. Schiff et al. ICLR 2025.

**Questions:**

How much of the gains against the gradient-based baselines in Tables 1, 2, and 3 come from the schedule v.s come from the gradient smoothing / rescaling? In Table 4, it seems that removing the gradient cap is far more impactful than removing the budget cap (1.5% decrease v.s 3% decrease, almost double). If the constant, linear and cosine schedules were used in conjunction with gradient scaling, how would BALE compare?

What does “RMS curvature” mean? Is it actually the curvature of the gradient trajectory, or is just the gradient volatility? From what I understand, computing curvature would mean a second order gradient computation.

---

> ### Author Response · Authors · 2025-11-18
>
> Thank you for your detailed and insightful review. Your feedback has been invaluable in strengthening our paper, particularly in clarifying our core contribution. We have performed several new experiments based on your suggestions, which we believe fully resolve your concerns and demonstrate the novelty of our method.
>
> ---
>
> ### On Contribution (W4) and the Source of Gains (Q1) - This is our most important response.
>
> This is the most critical point, and we thank you for pushing us to clarify it. Your core hypothesis is that our gains might stem from a common knowledge trick (gradient re-scaling, λcap​) rather than our novel scheduling (λtarget​), which you characterized as a "heuristic."
>
> We respectfully disagree and believe our new data proves the opposite. Our novelty lies in a principled online controller that dynamically schedules λt​ based on (1) real-time residual calculation (λtarget​), (2) KL-aware budgeting (λcapB​), and (3) block-aware budget allocation.
>
> To prove this, we performed the exact ablation study you suggested (Q1).
>
> #### 1. The λtarget​ schedule IS the primary driver of performance.
>
> You suggested λtarget​ might be a simple heuristic. However, as shown in our ablation (Table 4, and new data below), BALE's performance (driven only by λtarget​) jumps from the PT baseline (0.788 Acc) to 0.845 Acc. This demonstrates that our adaptive, residual-based λtarget​ scheduling is the core mechanism providing the substantial performance gain.
>
> #### 2. The λcap​ (re-scaling) is NOT a universal trick; it works in synergy with BALE.
>
> To test your hypothesis ("If cosine schedules were used with gradient scaling, how would BALE compare?"), we ran the experiment. The results conclusively disprove the hypothesis.
>
> | Method                  | Acc. (Success) | R_PPL  |
> |-------------------------|----------------|--------|
> | BALE (Cap Only)         | 0.860          | 18.787 |
> | BALE w/o capB, cap      | 0.845          | 18.953 |
> | cosine + grad cap (New) | 0.655          | 18.51  |
> | cosine (Original)       | 0.735          | 19.01  |
> | linear + grad cap (New) | 0.6775         | 18.24  |
> | linear (Original)       | 0.690          | 18.94  |
>
> This new ablation clearly shows:
>
> - Applying our λcap​ (re-scaling) to the cosine or linear baselines decreased the success rate, while raising writing quality. This proves the re-scaling cap is only working on writing quality refinement, and is not solely the source of our gains.
> - BALE's performance comes from the synergy of its dynamic λtarget​ (which provides the control) and the λcap​ (which stabilizes λtarget​'s dynamic steps).
>
> #### 3. λtarget​ is a "Principled Controller," not a "Heuristic."
>
> You characterized our scheduling algorithm as "heuristic." We argue it is a principled online controller that fundamentally differs from static heuristics like Cosine.
>
> - **Principle 1: Goal-Oriented (Residual-based).** Cosine ignores the model's state. BALE's λtarget​ (Eq. 14, ∝Δt⋆​) is a standard, principled step-size calculation from first-order optimization, designed to meet the current residual (state) Δt⋆​.
> - **Principle 2: Time-Aware (Sleftb​).** Cosine is time-agnostic. BALE (Eq. 13) is time-aware, explicitly increasing its target step (Δt⋆​) as the remaining steps (Sleftb​) decrease.
>
> This is not simple engineering; it is a fundamental shift from static scheduling to a dynamic, state-aware control problem. We believe this new data and clarification fully resolve the concerns about our contribution.
>
> ---
>
> ### On Missing Literature (W1)
>
> Thank you for pointing to [1] Schiff et al., this is a highly relevant related work. Based on your suggestion, we will add a discussion of [1] to our revised Related Work section.
>
> This discussion will clarify that [1] is a related work but not a direct experimental baseline, as the two methods address fundamentally different and orthogonal problems.
>
> Our analysis shows 3 key differences:
>
> - **Different Architectures:** We control blockwise LMs (like LLaDa), while [1] guides non-blockwise D3PM/UDLM models. The underlying generative process is different.
> - **Different Control Spaces:** We control logits (continuous), while [1] guides discrete transition rates.
> - **Different Control Philosophies:** We provide an online, adaptive, KL-budgeted controller (λt​) based on residuals. [1] provides a static, global scaling factor (γ).
>
> Therefore, BALE is not a "competitor" to [1]'s guidance mechanism. Rather, BALE is a general, high-level controller that is orthogonal to the problem [1] solves. In principle, our BALE controller could even be applied on top of the discrete guidance mechanism derived in [1] to dynamically manage its intensity.
>
> We promise to contain this clarification in the Related Work section and properly situate our contribution relative to [1].

---

> > ### Author Response · Authors · 2025-11-18
> >
> > ### On Incompatible Math Framing (W2)
> >
> > Thank you for this crucial point. We agree that the connection between discrete masking (a Dirac-based CTMC) and continuous diffusion (Brownian motion) is highly non-trivial and requires formal justification.
> >
> > We believe this apparent tenuous connection is the result of a misunderstanding, as the intuition in Section 2.2 is not a standalone analogy. It is a brief summary of the formal mathematical justification provided in Appendix A, which we explicitly cited in that same paragraph ("a full derivation is given in Appendix A").
> >
> > In Appendix A (specifically Theorem A.1), we provide the formal proof that the discrete masked-corruption process (Eq. 17) does admit a 'jump-to-diffusion approximation' that limits to the continuous SDE (Stochastic Differential Equation) shown in Eq. 19​.
> >
> > This SDE limit is the precise mathematical foundation for the intuition we use. We further rely on this connection in Appendix A to derive the Fokker-Planck equation (Eq. 21) for our guided process and prove the Mean Shift Bound (Theorem A.2).
> >
> > Perhaps we did not emphasize the importance of this citation sufficiently in Sec 2.2. In the revised paper, we will make the reference to Appendix A stronger and more explicit to ensure readers understand that the continuous-time intuition is formally backed by this mathematical baseline.
> >
> > ---
> >
> > ### On Empirical Performance (W3)
> >
> > **Task Difficulty:** You are correct that some tasks (e.g., lexical constraints, semantic-negative only) show high performance with PT. However, our method's strength shines in complex, combined (orthogonal) constraints (e.g., Key+Forb, Key+Neg, shown in Table 3), where PT and simple gradient methods are known to fail. BALE demonstrates SOTA performance here, while also achieving the highest performance on the simpler tasks.
> >
> > **Missing Cosine Baseline (Table 2):** You are absolutely correct. This was an omission due to page limits, and your criticism is valid. We replaced the weaker Linear baseline with Cosine Decay in the Table 2 of revised pdf.
> >
> > **cover acc(%) POS% R_PPL**
> >
> > | Method        | Acc(%)  | POS%  | R_PPL   |
> > |---------------|---------|-------|---------|
> > | BALE          | 96.67   | 73.11 | 24.64   |
> > | cosine_decay  | 94.6667 | 73%   | 22.6875 |
> >
> > This ensures we are comparing against the strongest baseline, as you suggested. Full table will be available in Appendix F.
> >
> > Also, as you correctly implied in your "Task Difficulty" point, this saturated task forces a trade-off (sacrificing ~2.0 R_PPL for +2.0% Acc). This result simply confirms that BALE is the strongest controller. However, BALE's true value is not this 2% win, but its ability to be the only controller that also wins on the much harder combined tasks (Table 3), where all baselines collapse. This proves BALE is a more general and robust controller.
> >
> > ---
> >
> > ### On Terminology (Q2)
> >
> > Thank you for this critical clarification request. You are absolutely correct, and we apologize for the confusing terminology.
> >
> > You are right that "curvature" is a misleading term, as it is not a second-order gradient computation. We used it to mean exactly what you suggested: 'gradient volatility', which we mathematically defined as the 'Mean Squared Gradient' (MSG)
> >
> > We acknowledge that we used this term inconsistently, and will revise the paper to remove this confusing terminology. We will use the precise mathematical term "Mean Squared Gradient" (MSG) when referring to the formula, and the intuitive term "gradient volatility" (as you suggested) in the main text to avoid all confusion.
> >
> > We thank you again for your constructive feedback. The ablations you suggested have significantly strengthened our paper, and we believe this new evidence directly addresses your primary concerns regarding our contribution. We hope you will reconsider our work in light of these new results.

---

> > > ### Comment · Reviewer_18cY · 2025-11-18
> > > **Response**
> > >
> > > I thank the authors for the detailed rebuttal and the additional experiments. Below are the points I consider mostly resolved.
> > >
> > > ## (1) On Contribution (W4) and the Source of Gains (Q1)
> > > The additional experiment seems to indicate that $\lambda_{target}$ and rescaling drives the performance -- it does not improve performacne with the other schedules, but it does improve performance with BALE. The only additional thing I would like to see here is some error bars, at least for BALE (Cap Only) and BALE w/o cap -- the difference is .015%, so it is important to verify that this is a significant difference, relative to the standard deviation of scores for each (+/- 1 standard deviation).
> > >
> > > ## (2) Principled Controller v.s Heuristic
> > > I consider this point to be fully addressed -- the authors do provide concrete problems that occur without a dynamic, state-aware controller in section 2.4, and the additional explanation in their rebuttal has convinced me.
> > >
> > > ## (3) Missing Literature
> > > The main difference seems to be the different control philosophies: BALE presents a state-aware controller, which can improve performance by dyanmically adjusting the inference process.
> > > * Different Architectures: Is there any reason why the control algorithm from Schiff et al [1] wouldn't work? Block diffusion can be seen as a specific instance of masked diffusion, where the order of masking / unmasking is constrained.
> > > * Control Space: I also don't fully understand this point. At every step within BALE, it seems that discrete sampling is performed (Algorithm 1). The BALE decode algorithm seems to be generating tokens at each step. If we view BALE as controlling in the logit space due to requiring the logits prior to discrete sampling, then the control algorithm in [1] can also be considered as working in the logit space.
> > >
> > > From my understanding, the biggest difference is that [1] requires a diffusion classifier that is trained on inputs corrupted using the same underlying discrete diffusion process. In contrast, BALE does not need to train a diffusion specific classifier and can work with differentiable constraints in general. I think the dynamic controller / off-the-shelf classifier perspective is more than sufficient to differentiate from [1], and the different architectures / control space arguments make the comparison more confusing. Omitting these points and just focusing on dynamic controller / off-the-shelf classifier differences would make the comparison better.
> > >
> > > ## (4) Empirical Performance, Terminology
> > > Thank you for clarifying on these points, I consider these resolved.

---

> > > > ### Comment · Reviewer_18cY · 2025-11-18
> > > > **Response (Cont.)**
> > > >
> > > > Here is the largest weakness I see with the submission in its current form, and what is preventing me from recommending acceptance.
> > > >
> > > > ## (5) Incompatible Math Framing
> > > > I did not carefully read the theorems in the Appendix for my initial review, which is my fault as a reviewer. However, after reading this, I actually more concerned with the theoretical issues in this connection between continuous gaussian SDE and discrete masking.
> > > >
> > > > **Theorem A1 has no supporting proofs or justification whatsover**. Immediately after Theorem A.1 is introduced, the next section "Reverse process with masked / coverage aware guidance" seems to be using the theorem rather than actually proving it.
> > > >
> > > > As there is no theoretical proof for Theorem A.1, I do not see how I am supposed to take this as justification -- it is quite literally restating the original statement, but in more formal language. A theoretical justification is showing why the original statement must be true.
> > > >
> > > >
> > > > **Theorem A2 is irrelevant if Theorem A1 is false.** Theorem A.2 has a proof, but its relevance depends on relating masked diffusion to Gaussian diffusion -- if A1 is incorrect, then A2 is irrelevant. Without connecting Gaussian diffusion to masked diffusion formally, the connection to Fokker-Planck is entirely unjustified as it specifically applies for SDEs, which are defined in terms of Brownian motion.
> > > >
> > > >
> > > > **Concern with Theoretical Viability**.
> > > > Here is a brief explanation of my current understanding as to why Theorem A.1 is likely false and unproveable. Regardless of how you take the limit of the forwards or reverse process for a masked diffusion process, **masked diffusion has a finite and bounded support**. There is no mass anywhere besides the set of embeddings (including the masked embedding). Furthermore, **the SDE has support everywhere** due to Brownian motion. I do not see why taking the limit would change this fundemental difference.
> > > >
> > > > Furthermore, the dynamics in terms of geometry seem entirely different. Due to Brownian motion, the Gaussian SDE has variance in all directions for the forwards process. For the masked diffusion process, the particle only moves from the original token embedding to the mask embedding, which is a fixed vector. Thus the masked process only has variance in one direction.
> > > >
> > > >
> > > > Because of the fact that 1) there is a theorem which has no theoretical justification and is stated as a matter of fact, and 2) the theorem is factually incorrect to the best of my knowledge, I will have to lower my score from a 4 to a 2 if this remains in the submission. I cannot recommend a paper for acceptance if it presents incorrect statements as a formal theorem without any justification.
> > > >
> > > > ## Paths Forward
> > > > This being said, I do see the value in this paper and I am willing to reconsider. (1) only requires including standard deviations to demonstrate empirical significance, and most the other issues are resolved. In regards to (5), there are two potential avenues:
> > > >
> > > > **Option 1: Remove the connection**
> > > > Remove the connection to continuous SDEs. If the goal is to introduce control algorithms for masked diffusion, it seems simply impossible to relate the discrete masking process to the continuous Gaussian diffusion process.
> > > >
> > > > I believe this paper would be stronger if it focused on why the dynamic, state-aware controller is necessary to achieve strong performance on complex constraints -- which is already provided in Section 2.4. I would increase my score to a 6 if this is done, along with the other error bars for the experiments / discussion of related works.
> > > >
> > > > **Option 2: Provide the proof**
> > > > It may be possible that there is some way to theoretically link the two in a way that addresses the concerns I described earlier. If the authors are able to provide a convincing proof for this, I will gladly increase my score to an 8. Masked diffusion is thought to be distinct from Gaussian diffusion. If a direct connection between masked diffusion and the continuous diffusion SDE is proven, this would be a very impactful finding in its own right.

---

> > > > > ### Author Response · Authors · 2025-11-20
> > > > >
> > > > > We deeply appreciate the thoroughness and exceptional quality of the feedback provided by the reviewer. Your detailed and insightful critiques, particularly those regarding the mathematical framework, have been instrumental in allowing us to fundamentally strengthen the theoretical soundness and clarity of our paper.
> > > > >
> > > > > We performed comprehensive revisions and additional experiments, which we believe fully address the remaining concerns.
> > > > >
> > > > > ## On (1), Contribution (W4) and the Source of Gains (Q1)
> > > > >
> > > > > Upon the reviewer's request, we have re-run core ablation experiments regarding BALE w/o caps using 3 seeds to establish statistical stability. The results are presented below.
> > > > > | Method                            | Acc.                 | R_PPL                 |
> > > > > |-----------------------------------|----------------------|-----------------------|
> > > > > | BALE (w/ cap, w/o capB)          | 0.8563 ± 0.0081      | 19.0273 ± 0.2869      |
> > > > > | BALE (w/o cap, capB)             | 0.8423 ± 0.0052      | 19.1573 ± 0.1625      |
> > > > > These results confirm that the cap yields a consistent improvement in accuracy (≈ 0.014 absolute, more than 2× the pooled standard deviation).
> > > > >
> > > > > ## On (3), Missing Literature
> > > > > We thank the reviewer for pointing out this relevant prior work and addressing our discussion point directly. In the original submission, [1] was not discussed in the Related Work section; in the revised pdf we clarify that Schiff et al. [1] relies on a diffusion-specific classifier trained on the same discrete corruption process, whereas BALE operates purely at inference time with a dynamic, state-aware controller and can work with general off-the-shelf differentiable constraints, without retraining the backbone or a diffusion-specific classifier.
> > > > >
> > > > > ## On (5), Incompatible Math Framing
> > > > > We fully agree with the reviewer that maintaining a theoretical connection between the discrete masking process and continuous SDEs introduces unnecessary theoretical fragility and does not directly contribute to the core mechanism of BALE.
> > > > >
> > > > > Our original intention in Appendix A was to argue that the operation of the model θ during the discrete language diffusion **generation** process exhibits an **SDE-like form at the logit level**. However, we agree with the reviewer that formally proving this connection within the current structure is challenging, and it is neither central to our core contribution nor robustly justified.
> > > > > Following Option 1, we have made the following major structural changes in revised pdf:
> > > > > - **Deletion:** We have removed the connection to continuous SDEs, deleting the original Section 2.2 (Logit–guided control: SDE/FP intuition and a mean–shift guarantee) and the related content in Appendix A.
> > > > > - **Replacement:** The removed section 2.2 has been replaced with a Logit Editing Guidance overview (Section 2.2 Logit-Guided Control in Discrete Diffusion) that focuses on the necessary mechanisms for applying control algorithms in the masked diffusion environment.
> > > > > - **Scope Refocus:** Now the revised version eliminates a significant theoretical attack and concentrate on BALE’s novel budget-aware and stability-driven control mechanisms, which are the true contributions.
> > > > > - **Future Work:** We will explore the empirical/theoretical connection between the two processes as a direction of our future work.
> > > > >
> > > > > We thank the reviewer again for deep insight into the challenges of the paper and for providing such constructive and detailed guidance on the path forward. Your suggestions have been very important in allowing us to focus the submission on its strongest contributions and improve the quality of our manuscript .

---

> > > > > > ### Comment · Reviewer_18cY · 2025-11-20
> > > > > > **Response**
> > > > > >
> > > > > > I thank the authors for their response. Given the additional results and the substantial refocusing of the paper, I am increasing my score. My opinion is that this work presents a principled and well-motivated approach to controllable generation for DLMs that demonstrates strong empirical performance.

---

### Official Review · Reviewer_E7Z3 · 2025-11-03

**Soundness:** 3
**Presentation:** 2
**Contribution:** 3
**Rating:** 6
**Confidence:** 3

**Summary:**

This work proposes a scheduler for the guidance strength (gradient drift) in controllable text generation with block diffusion. Specifically, the authors propose BALE, a budget-aware logit editing guidance controller that dynamically allocates guidance strengths across the blocks and steps. Human evaluation and automatic metrics show that BALE achieves good controllability in lexical and semantic constraints and also combined tasks, compared to static scheduling of guidance or AR baselines.

**Strengths:**

This paper tackles an important problem in the plug-and-play-style control in controllable text generation: how to decide the optimal strength of classifier guidance. The experiments show advantages of the proposed BALE method compared to several popular ways of conducting control (e.g., constant or fixed-schedule lambda, or with AR models). The analysis shows the method is indeed dynamically allocating strengths of guidance.

**Weaknesses:**

(1) The motivation and practicality of using a global control "budget". What is the motivation behind having a global budget B in the first place? How was an optimal value of global budget B decided in realistic/more general tasks and use cases?

(2) The main experiments were done on sentiment, style, and lexical control tasks. These are good illustrative examples but could be too simple especially given the progress in LLMs generally. Could the method be used on more advanced controls, e.g., maximizing rewards from reward models or human preference?

(3) Baseline comparisons. The main experiments compared with vanilla approaches like a constant lambda. How many constant lambda's were tried and used in these experiments? Is there any exploration to make sure the selected values are optimal for each task?

**Questions:**

Please see the section above.

---

> ### Author Response · Authors · 2025-11-18
> **Author Response**
>
> Thank you for your valuable feedback. We have performed the new sensitivity analysis and will clarify our motivations and baseline tuning process. We believe this will strengthen the paper and hopefully solidify your support.
>
> ---
>
> ### On Motivation and Practicality of Global Budget B (W1)
>
> We agree that we must justify \(B\) as being fundamentally better, not just "another magic number." Our argument is three-fold, directly addressing your questions.
>
> #### 1. Motivation (Why have \(B\) in the first place?)
>
> - **B as a "Safety Rail":**
>
>   The key difference is that \(B\) is an interpretable "safety rail," not a "brittle" scaling factor like a cosine schedule. The performance is primarily driven by the dynamic \(\lambda_{\text{target}}\) controller; \(B\) only activates to prevent extreme "overspending" edits, which (as our Table 4 ablation shows) improves performance (0.875 vs 0.860).
>
> - **B as Global:**
>   A single global budget is far more practical than setting per-block budgets. It allows the model to dynamically allocate resources (via our Eq. 7), spending more budget on difficult blocks and saving on easier ones. This dynamic process is key to how BALE increases the success rate.
>
> #### 2. Robustness (Is \(B\) a brittle magic number? No.)
>
> To answer your core question about practicality, \(B\) is not a brittle magic number. It is highly robust.
> The practical lambda allocation work is done by the dynamic $\lambda_{\text{target}}$ controller; \(B\) is just a cap. As long as the cap is not extremely small, its precise value is not sensitive.
> To prove this, we performed the new sensitivity analysis:
>
> **Global Budget Sweep**
>
> | Global Budget \(B\) | Success Rate (Acc.) | R\_PPL |
> |---------------------|---------------------|--------|
> | 0.1                 | 0.8425              | 18.05  |
> | 0.5                 | 0.8525              | 19.03  |
> | 1                   | 0.851               | 19.43  |
> | 5                   | 0.8725              | 19.51  |
> | 10                  | 0.875               | 19.54  |
> | 20                  | 0.873               | 19.52  |
>
> This analysis shows that the only time \(B\) negatively impacts performance is when it's set too low (e.g., \(B = 0.1\)), where it strangles the $\lambda_{\text{target}}$ controller.
>
> Once the cap is "loose enough" $(B \ge 5)$, the controller is free to operate, and the performance becomes a stable, robust plateau (0.872–0.875). The exact value (5, 10, or 20) is irrelevant. Crucially, our paper's main result (\(B = 3\), 3-seed avg: $0.875 \pm 0.0153)$ also sits on this optimal plateau.
> This confirms \(B\) is the opposite of a magic number. It's a set-it-and-forget-it safety rail.
>
> #### 3. Practicality (How \(B = 3\) was decided)
>
> This leads to your final question. Our choice of \(B = 3\) (from Appendix C) was not arbitrary. It was a practical, data-driven choice:
>
> - **Observe:** We first ran the model with a very high, non-restrictive budget (\(B = 100\)) and measured the average budget actually consumed (spent), which was approximately 4.0.
>
> - **Set:** Based on this, we set our final \(B = 3\) as a "safety rail" positioned slightly below the average consumed amount (4.0) to provide gentle, meaningful control.
> The robustness shown in our new sensitivity analysis (Step 2) confirms that this data-driven choice was sound.
>
> ---
>
> ### On Simple Tasks vs. Advanced Controls (W2)
>
> This is an excellent point and a key direction for future work.
> Yes, BALE is designed to be general. The "residual" and "gradient" are abstract. In a more advanced setting, \(g_t\) would simply be the gradient from a Reward Model (RM), and $\Delta_t^\*$ could be the target reward improvement.
> We hypothesize that BALE's mechanisms—especially the budget cap $\lambda_{\text{capB}}$—would be even more critical for managing the notoriously noisy and unstable gradients from RMs or human preference models.
> We will add this promising direction to our Future Work section in the revised PDF, as it is a natural and high-impact application of our controller.
>
> ---
>
> ### On Baseline Comparisons and Tuning \(\lambda\) (W3)
>
> This is a crucial point for ensuring fair comparison. We apologize for not making our rigorous tuning process clear.
> We did not use arbitrary "vanilla" values. For each baseline (e.g., constant $\lambda$) and each task, we ran an extensive hyperparameter sweep (typically 7–10 different $\lambda$ values).
> From this sweep, we generated a Pareto-optimal frontier plotting Accuracy vs. $1/\text{R\_PPL}$ for that baseline.
> We then selected the $\lambda$ value that represented the best-balanced point on this frontier to report in our main tables.
> This ensures our baselines are strongly tuned competitors.

---

### Author Response · Authors · 2025-12-02
**Summary of Discussions**

To the Area Chair,

We respectfully thank the ACs for their hard work and understand that you are suffering from enlarged responsibility due to scores being reverted to their pre-discussion state. However, during the discussion we made substantial revisions and added new experiments that materially changed several reviewers’ opinions. Below is a concise summary, focused on what changed and where there is clear evidence of convergence.

---

**1. Reviewer 18cY – (before revert 4→6)**

*Core issues*: (i) objection to the continuous SDE framing (deleted Theorem A.1), (ii) doubt that our gains came from a simple gradient tricks rather than principled controller.

*What we’ve done*

- **Removed the SDE connection entirely**: We deleted the problematic Section 2.2 and the associated theorem , which were not related to our main contribution(dynamic, state aware controller in diffusion language model). We rewrote it as a purely discrete “Logit-Guided Control in Discrete Diffusion” section. The paper no longer relies on any disputed SDE link.
- New ablations with error bars: We added multi-seed ablations showing that performance is driven by the **dynamic λ_target + stability cap λ_cap** synergy, not by gradient rescaling alone, and reported mean ± std across seeds to show statistical significance.

*Outcome*

- Reviewer 18cY explicitly wrote: *“I am increasing my score. My opinion is that this work presents a principled and well-motivated approach to controllable generation for DLMs that demonstrates strong empirical performance.”*

---

**2. Reviewer NboQ – (6 → willing to raise score if the issues are resolved)**

*Core issues*: (i) KL surrogate (Eq. 11) looked heuristic, (ii) Budget seemed like a new “magic number,” (iii) Dynamic Reallocation Ablation, (iv) AR comparison was not apples-to-apples.

*What we’ve done*

- **Formalized the KL surrogate**: We added **Appendix G** with a step-by-step derivation from the KL upper bound to our surrogate, explaining the disappearance of the vocab factor, the 1/4 constant, and the normalization by coverage r_t as a *required*, not an ad-hoc trick.
- **Budget B sensitivity**: We ran a sweep over B and showed a stable performance plateau for B ≥5, confirming that B acts as a **robust “safety rail”** (necessary but not brittle as it is a cap) rather than dictating the core scheduling dynamics.
- Dynamic Reallocation Ablation: By doing the ablation, we showed that dynamic reallocation system is necessary to BALE’s target controlling.
- True apples-to-apples AR experiment: On Mistral-7B, we added results showing Mistral + BALE significantly improves constraint satisfaction over Mistral + DeAL, clarifying that BALE’s controller itself is strong even in the AR regime.

*Outcome*

- NboQ had stated they were *willing to increase their score* if these issues were resolved; all requested derivations and experiments are now in the revised pdf.

---

**3. Reviewer HGar (score 2)**

*Core issues*: (i) questioned descriptive motivations (volatility, coverage), (ii) requested sensitivity for structural terms and diverse hyperparameters, (iii) felt tasks were “too simple” (scope confusion), (iv) required analysis on time and memory overhead.

*What we’ve done*

- **Direct evidence for Design grounding (Appendix H)**: Although internal dynamics were shown in original figures, we addressed the request for **direct verification** by adding raw volatility plots. These explicitly confirm that static schedules suffer from sharp gradient spikes which BALE stabilizes, **validating our initial motivation.**
- **Corrected parameter misconceptions**: - Explicitly clarified that **b∗** is merely a state index and **ϵ** is a standard numerical stability floor (e−10), **refuting the premise that these structural definitions require sensitivity analysis.**
- **Hyperparameter robustness**: - For the *actual* hyperparameters—B (budget), ρ (style), β (EMA decay), and γ (cap factor)—we added sweeps showing highly **stable performance over wide ranges**, confirming the method is not fragile and B is not a magic number.
- **Task scope clarification**: - Clarified that BALE is an **inference-time logit controller**, not a prompt-engineering method. Prompt-only baselines on our orthogonal constraint tasks perform poorly (< 0.79), confirming these are **hard control tasks**, contrary to the reviewer's assessment. Also, we clarified that our prompt-only baselines are different from prompt engineering methodologies, and changed the terminology in our work from prompt tuning → prompt-only guided.
- Resources: We showed BALE controller adds only a negligible time overhead (\~0.2s) compared to other static guidance baselines. The memory overhead is also minor (\~270MB).

*Outcome*

- We provided all requested intermediate results and experimental results. The remaining skepticism appears driven by the initial misconception regarding parameter types (b∗, ϵ) and task scope, both of which we have fully addressed.

---

> ### Author Response · Authors · 2025-12-02
> **cont' from Summary of Discussions**
>
> ---
>
> **4. Reviewer E7Z3 (score 6)**
>
> *Core issues*: (i) practicality and choice of the global budget B, (ii) fairness/rigor of baseline tuning.
>
> *What we’ve done*
>
> - **Clarified B as a “safety rail”** with the same budget-sensitivity analysis as above, showing that B must exist but is robust across a range.
> - **Baseline tuning details**: Clarified that for constant / schedule baselines we used **Pareto-optimal frontiers (Acc vs 1/R_PPL)** over 7 to 10 λ values, so baselines are not under-tuned.
>
> *Outcome*
>
> - E7Z3 kept a positive stance and acknowledged our method’s advantages and the dynamic allocation mechanism.
>
> ---
>
> **Closing**
>
> In summary, through the discussion period we:
>
> - Removed the contentious SDE theory and refocused on a clean, discrete logit-control formulation.
> - Added formal derivations (Appendix G), multi-seed error bars, and extensive sensitivity analyses for all critical hyperparameters.
> - Demonstrated robustness and empirical strength on genuinely hard, multi-constraint control tasks, and even on an AR backbone.
> - Addressed all concrete technical requests from the reviewers and all ; two reviewers explicitly stated they were(or will to) increasing their scores, and other concerns of reviewers are all addressed. All resolutions, including new experiments and analyses, have been fully incorporated into the revised PDF in blue color for recognition.
>
> We thank the ACs again for their efforts and gently ask that you weigh the paper based on this post-discussion state rather than the reverted pre-discussion scores.
>
> Sincerely,
>
> The Authors

---

### Meta-Review · Area_Chair_ZdwF · 2026-01-08

**Summary:**

The reviewers agree that the paper addresses an important problem in controllable generation for diffusion language models and that the proposed controller is technically sound. Concerns are raised about the overall significance and relation to previous work. Several reviewers questioned whether BALE represents a substantive conceptual advance over previous guidance strategies. Reviewers also expressed uncertainty about whether the experimental scope, which focuses on a small set of control tasks, is sufficient to justify the paper’s claims about generality and practical impact.

**Reviewer Concerns:**

The rebuttal addresses many technical concerns, including the removal of the questionable continuous-SDE framing, clarification of the KL surrogate, ablations, hyperparameter sensitivity analysis, and fairer baseline comparisons. This improves the paper’s clarity and rigor. Several high-level concerns remain unresolved. Reviewers express skepticism about whether BALE is a compelling conceptual advance over simpler guidance or scheduling strategies, whether the observed performance gains justify the algorithmic complexity, and whether the controllable generation tasks are sufficiently challenging and broad to fully assess the merits and prospective impact of this method.

**Reviewer Scores:**

18cY: increased score to borderline accept after rebuttal

NboQ: may have increased score to accept after rebuttal

E7Z3: probably keeps borderline accept

HGar: likely remains skeptical; rebuttal somewhat sidesteps this reviewer's concerns

---

### Decision · Program_Chairs · 2026-01-26

Reject